# The long non-coding RNA *MIR31HG* regulates the senescence associated secretory phenotype

Marta Montes [1 ✉], Michal Lubas[1], Frederic S. Arendrup [1], Bettina Mentz[1], Neha Rohatgi[2], Sarunas Tumas[1,4], Lea M. Harder[3], Anders J. Skanderup [2], Jens S. Andersen [3] & Anders H. Lund [1 ✉]

Oncogene-induced senescence provides a barrier against malignant transformation. However, it can also promote cancer through the secretion of a plethora of factors released by senescent cells, called the senescence associated secretory phenotype (SASP). We have previously shown that in proliferating cells, nuclear lncRNA *MIR31HG* inhibits p16/CDKN2A expression through interaction with polycomb repressor complexes and that during BRAF-induced senescence, *MIR31HG* is overexpressed and translocates to the cytoplasm. Here, we show that *MIR31HG* regulates the expression and secretion of a subset of SASP components during BRAF-induced senescence. The SASP secreted from senescent cells depleted for *MIR31HG* fails to induce paracrine invasion without affecting the growth inhibitory effect. Mechanistically, *MIR31HG* interacts with YBX1 facilitating its phosphorylation at serine 102 (p-YBX1$^{S102}$) by the kinase RSK. p-YBX1$^{S102}$ induces *IL1A* translation which activates the transcription of the other SASP mRNAs. Our results suggest a dual role for *MIR31HG* in senescence depending on its localization and points to the lncRNA as a potential therapeutic target in the treatment of senescence-related pathologies.

[1] Biotech Research and Innovation Centre, University of Copenhagen, Copenhagen, Denmark. [2] Genome Institute of Singapore, Agency for Science, Technology and Research (A*STAR), Singapore, Singapore. [3] Department of Biochemistry and Molecular Biology, University of Southern Denmark, Odense, Denmark. [4] Present address: Novo Nordisk Foundation Center for Biosustainability, Technical University of Denmark, Lyngby, Denmark. ✉email: marta.montes@bric.ku.dk; anders.lund@bric.ku.dk

Cellular senescence is an irreversible state of growth arrest that can be driven by several stimuli including telomere shortening due to extensive replication, DNA damage, oxidative stress or oncogene overexpression[1,2]. In addition, it has been recently shown that senescence can play a role in differentiation and tissue regeneration[3,4]. Oncogene-induced senescence (OIS) was firstly reported by Serrano et al.[5] when they observed that expressing an oncogenic form of Ras in primary fibroblastinduced senescence. Later work demonstrated that other oncogenes such as mutated BRAF promoted OIS both in vivo and in vitro[6,7]. OIS has been considered as a barrier to prevent tumour progression and additional mutations in tumour suppressor genes are required in order to bypass senescence to promote malignancy[7–9]. Senescent cells show a characteristic morphology and biochemical features such as halted proliferation, enlarged size, activation of the senescence-associated β-galactosidase, expression of cell cycle inhibitors and the presence of senescence-associated heterochromatin foci. Importantly, in several types of senescence, the cells secrete factors such as interleukins, cytokines and metalloproteases, which are part of the senescence-associated secretory phenotype (SASP) that can impact the cellular environment and homoeostasis of the neighbouring tissues[10–12]. The downstream effects of the SASP can be beneficial or detrimental depending on the tissue context[13,14]. It has been shown to prevent cancer progression by reinforcing autocrine senescence, inducing paracrine senescence in neighbouring cells[11,12], and by inducing tissue repair and regeneration[3,4,15]. The SASP can also activate the immune system facilitating the clearance of damaged cells[16,17]. On the other hand, the SASP can promote tumorigenic processes such as angiogenesis and invasion[10,18]. During aging, excessive SASP secretion can induce chronic inflammation that can lead to aged-related pathologies[19,20]. The composition of the SASP is very variable depending on different aspects such as the senescence stimuli and the cell type[21,22]. Defining the composition of the secretome and identifying new regulators in each biological context is crucial to identify molecular signatures of such a complex phenotype.

Previously, the long non-coding RNA (lncRNA) TERRA has been shown to be a component of inflammatory exosomes and to modulate transcription of inflammatory cytokines in recipient cells[23] and other lncRNAs have been reported to regulate nuclear factor κB (NF-κB) activation and its downstream target genes[24–26]. These findings raise the hypothesis that lncRNAs could act as key players in the SASP induction during senescence. A few lncRNAs have been directly linked to senescence[27–29]. We have previously identified the lncRNA MIR31HG to be upregulated in BRAF mediated OIS[28]. In proliferating cells nuclear MIR31HG represses p16/CDKN2A expression by recruiting Polycomb group complexes. Indeed, we have shown that in melanoma patients harbouring BRAF mutations, MIR31HG negatively correlates with p16/CDKN2A expression[28]. Upon BRAF induction MIR31HG is upregulated and translocates to the cytoplasm where we now show that it regulates the expression of a part of the SASP repertoire. MIR31HG knock-down in BRAF-induced senescent fibroblasts reduces expression and secretion of several components of the SASP. Interestingly, conditioned medium (CM) from MIR31HG-depleted senescent cells promotes paracrine senescence but fails to induce cancer cell invasion. Mechanistically, we show that MIR31HG promotes the interaction between YBX1 and the RSK kinase resulting in YBX1 phosphorylation and subsequent induction of IL1A translation. Our results unveil the role of lncRNAs in the regulation of the SASP highlighting the dual role in senescence of MIR31HG by suppressing CDKN2A expression in young cells and facilitating production of a distinct subset of SASP factors in senescent cells.

## Results

**MIR31HG depletion affects SASP induction during OIS.** To assess the role of MIR31HG during OIS we used immortalized human fibroblasts BJ-hTERT, expressing a constitutively active form of the mouse B-RAF (V600E) that is fused to the oestrogen receptor (BJ ER:BRAF). Addition of 1 μM of 4-hydroxitamoxifen (4-OHT) activates BRAF inducing OIS[30]. We first confirmed that MIR31HG was overexpressed in BJ ER:BRAF upon 4-OHT induction (Fig. 1a), as we previously reported for TIG3 ER:BRAF cells[28]. In validation of the model, analysis of TCGA data from thyroid and colorectal cancer tumour samples, where BRAF is frequently mutated, demonstrated a higher MIR31HG expression in tumours harbouring BRAF mutations (Fig. 1b). We performed RNA sequencing in control cells and BRAF-induced senescent cells transfected with control siRNA or siRNA targeting MIR31HG. As expected, many genes were differentially expressed comparing control versus senescent cells (Fig. 1c and Supplementary Data 1). Moreover, the transcriptional profile of MIR31HG knock-down senescent cells resembled the senescent profile with several clusters of differentially expressed genes (Fig. 1c and Supplementary Data 1). Among the GO categories significantly represented on the genes downregulated in the MIR31HG knock-down conditions we found cytokine and chemokine-related pathways (Supplementary Fig. 1a). Furthermore, we observed that senescent MIR31HG-depleted cells failed to upregulate part of the main SASP components previously defined[13] (Fig. 1d and Supplementary Data 1). We validated transcriptomics data by quantitative real-time reverse transcriptase PCR (RT-qPCR) and confirmed the RNA decreased on several interleukins and chemokines such as interleukin-6 (IL6) and CXCL1, whereas other factors such as ICAM1 or IL1A were not affected (Fig. 1e). To validate our results in another cellular system, we performed RT-qPCR analysis in TIG3 ER:BRAF cell line. Depletion of MIR31HG during BRAF-induced senescence decreases the levels of the SASP RNAs (Supplementary Fig. 1b). To investigate a broader effect of MIR31HG in another senescence model, we assessed doxorubicin-induced senescence in BJ cells. As expected, cells showed senescence markers after 48 h treatment with 500 nM doxorubicin as seen by a reduced cell growth (Supplementary Fig. 1c) and the presence of β-galactosidase-positive cells (Supplementary Fig. 1d). Doxorubicin induced the expression of MIR31HG and also SASP components such as IL6, IL8 or CXCL1 (Supplementary Fig. 1e). p21 mRNA was also induced consistent with decreased cell proliferation (Supplementary Fig. 1e). Interestingly, depletion of MIR31HG with two different siRNAs resulted in decreased expression of SASP components but not p21 (Supplementary Fig. 1e).

In order to analyse the composition of the SASP at the protein level, we knocked down MIR31HG in BJ ER:BRAF cells and induced senescence by 72 h 4-OHT treatment in serum-free medium followed by mass spectrometry analysis of secreted proteins. Untreated cells transfected with control siRNA were used as control. Control senescent cells, when compared to control proliferating cells, showed differential secretion of factors (Supplementary Fig. 1f and Supplementary Data 2). The secretome of senescent MIR31HG knock-down cells resembled that of control senescent cells with discrete differences (Fig. 1f and Supplementary Fig. 1f and Supplementary Data 2). Among the less secreted proteins upon MIR31HG depletion were inflammatory SASP factors IL6 and CXCL1 (Fig. 1f and Supplementary Fig. 1f–h). The reduced secretion of these factors and other components of the SASP (IL8 and MMP3) were validated by enzyme-linked immunosorbent assay (ELISA) and western blotting using two different siRNAs for MIR31HG (Fig. 1g and Supplementary Fig. 1i). Altogether our results demonstrate that MIR31HG

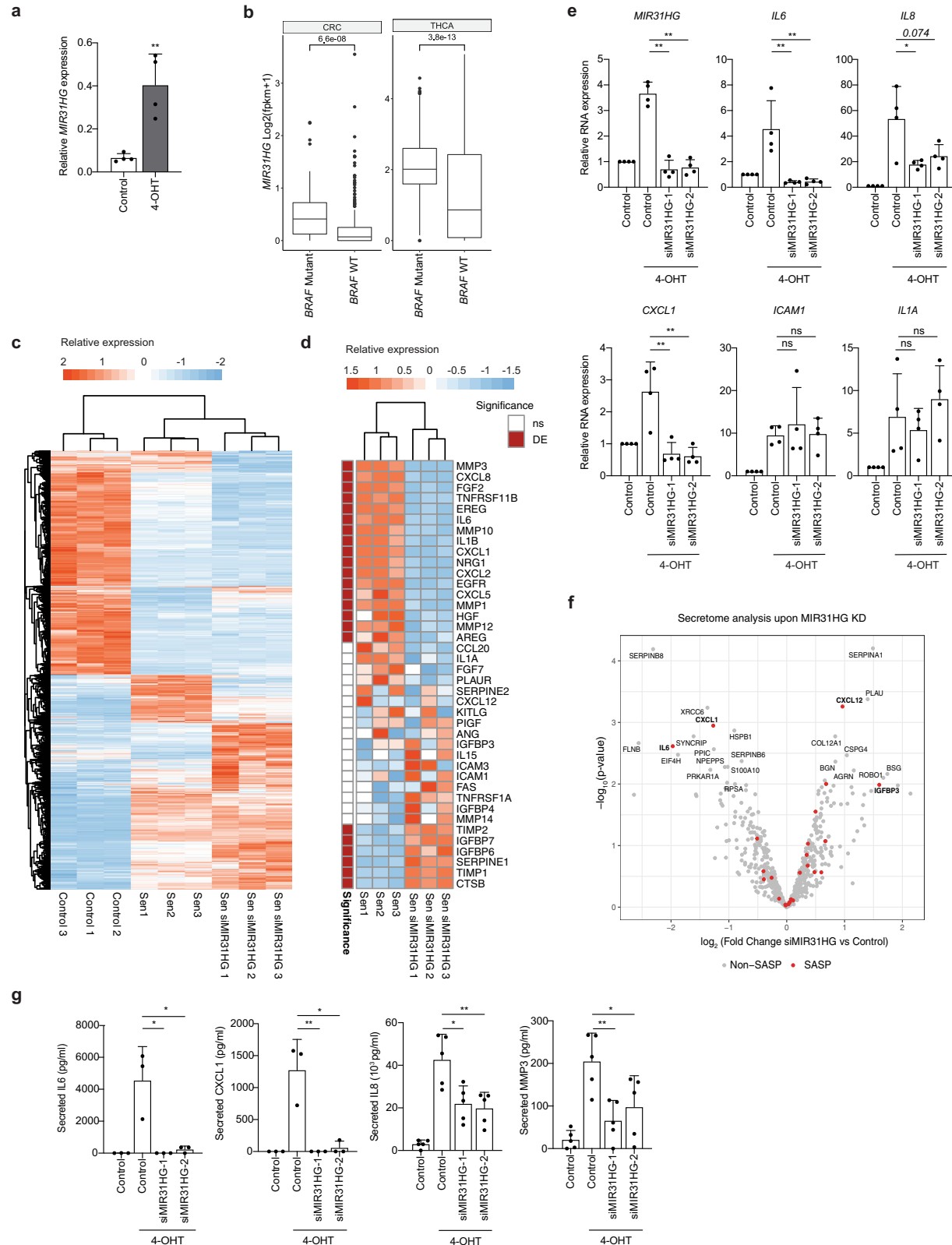

knock-down reduces the production of several SASP components during OIS already at the transcriptional level and these changes are reflected in the secretome.

**Paracrine effect of the SASP of *MIR31HG*-depleted cells**. To determine the paracrine effect of the SASP we induced senescence in cells transfected with control siRNA or siRNAs targeting

*MIR31HG*. After 72 h 4-OHT treatment CM was harvested and used for subsequent analysis. The CM from proliferating cells without 4-OHT induction was also harvested at the same time point as control. To assess the potential of the SASP in inducing paracrine senescence, we incubated BJ and IMR90 WT cells with the different CM for 72 h. Protein analysis of whole-cell extracts by western blot revealed that BJ and IMR90 cells growing in CM

**Fig. 1 MIR31HG knock-down decreases the induction of SASP components during BRAF-induced senescence. a** Relative *MIR31HG* expression normalized to housekeeping genes (*HPRT1* and *RPLP0*) in BJ ER:BRAF cells were treated with ethanol (Control) or 1 µM 4-OHT for 48 h (*n* = 4). **b** Box plot showing *MIR31HG* expression in log2 (fpkm + 1) units in thyroid carcinoma (THCA) and colorectal carcinoma (CRC) comparing BRAF mutant and BRAF wild-type tumours. The box plot represents the median (middle line), the box indicates the first and third quartiles and the whiskers indicate ±1.5 × interquartile range. Two-tailed Wilcoxon test was performed to compare the expression differences in the BRAF mutant and BRAF wild-type tumours (CRC BRAF_wt = 309; CRC BRAF_mutant = 49; THCA BRAF_mutant = 290; THCA BRAF_wt = 199). **c** Heat map showing relative expression of differentially expressed genes in BJ ER:BRAF cells (control or siMIR31HG), treated with ethanol (Control 1–3) or 1 µM 4-OHT for 48 h (Sen 1–3 and Sen siMIR31HG 1–3). **d** Heat map showing relative expression in RPKMs of a subset of SASP genes previously defined[13] from the data provide in **a**, where differentially expressed genes ('DE', FDR < 0.01) are indicated in red, unchanged (non-significant, 'ns') in white. **e** qRT-PCR analysis of selected components of the SASP normalized to housekeeping genes (*HPRT1* and *RPLP0*) in BJ ER:BRAF cells transfected with the indicated siRNAs (Control or siMIR31HG1-2), treated with ethanol (Control) or 1 µM 4-OHT for 48 h. The graphs show results compared to control ethanol-treated set to 1 (*n* = 4). **f** Mass spectrometry-based secretome analysis of senescent BJ ER:BRAF (1 µM 4-OHT) treated with siMIR31HG or control siRNA. The volcano plot shows differentially secreted proteins in *MIR31HG* knock-down senescent cells (Sen siMIR31HG) compared to control senescence cells (Sen) (−log10 *p* value along *y*-axis and log fold change along *x*-axis). Names are displayed for significantly changed proteins between secretomes (*pval* < 0.01) and SASP proteins are marked in red. **g** Secreted IL6, CXCL1, IL8 and MMP3 (pg/ml) measured by ELISA in the CM of BJ ER:BRAF (Control or siMIR31HG1-2) treated for 72 h with ethanol (Control) or 1 µM 4-OHT (*n* = 3–5). All statistical significances were calculated using two-tailed Student's *t*-tests, except **b** where two-tailed Wilcoxon test was used. *$p < 0.05$; **$p < 0.01$; ***$p < 0.001$; ns non-significant. All error bars represent means ± s.d. Source data are provided as a Source Data file.

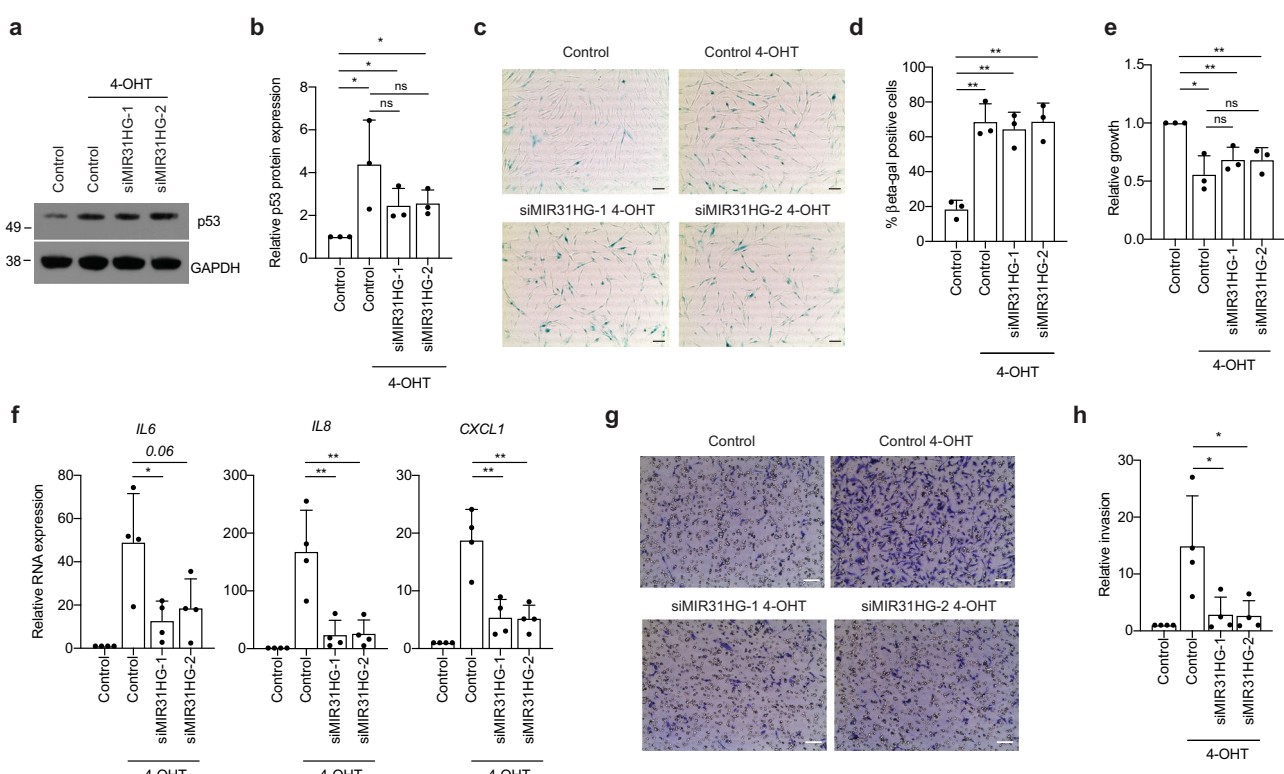

**Fig. 2 The SASP of senescent MIR31HG knock-down cells induces senescence but not invasion in a non-autonomous manner. a** Western blot analysis for p53 and GAPDH. in WT BJ cells incubated for 72 h with the CM collected from BJ ER:BRAF cells (Control or siMIR31HG1-2) treated with ethanol (Control) or 1 µM 4-OHT for 72 h. Molecular weight marker is shown in kDa. **b** Quantification of the western blot band intensities for p53 relative to GAPDH (*n* = 3). **c** Representative images of β-galactosidase staining from the same condition as indicated in **a** (*n* = 3). Scale bar: 50 µm. **d** Quantifiaction of the% of β-galactosidase-positive cells from **c** (*n* = 3). **e** Relative growth by crystal violet staining dissolved in acetic acid and measured at 590 nm in WT BJ cells incubated for 72 h with the CM collected from BJ ER:BRAF cells (Control or siMIR31HG1-2) treated with ethanol (Control) or 1 µM 4-OHT for 72 h. The graph shows the absorbance relative to the control cells set as 1 (*n* = 3). **f** qRT-PCR analysis of a subset of components of the SASP normalized to housekeeping genes (*HPRT1* and *RPLP0*) in total RNA extracted from cells described in **a**. The graph shows the RNA expression relative to control ethanol-treated cells set to 1 (*n* = 4). **g** MDA-MB-231 invading cells through a matrigel membrane in contact with the CM collected from BJ ER:BRAF cells (Control or siMIR31HG) treated for 72 h with ethanol (Control) or 1 µM 4-OHT. Representative images are shown in the figure (*n* = 4). Scale bar: 50 µm. **h** Quantification of the invading cells relative to control ethanol-treated cells (*n* = 3). All statistical significances were calculated using two-tailed Student *t*-tests, *$p < 0.05$; **$p < 0.01$, ns, non-significant. All error bars represent means ± s.d. Source data are provided as a Source Data file.

from control senescent cells and *MIR31HG* knock-down senescent cells were able to mildly upregulate the senescence marker p53 (Fig. 2a, b and Supplementary Fig. 2a) as well as β-galactosidase (Fig. 2c, d) compared to proliferating cells. Consistently, a subset of p53 target genes[31] was upregulated in BJ and

IMR90 cultured in CM from senescent cells in the presence and absence of *MIR31HG* (Supplementary Fig. 2b, c). Moreover, cells growing in CM from senescent cells showed decreased cell number after 72 h, independently on *MIR31HG* expression (Fig. 2e and Supplementary Fig. 2d). This was consistent with

decreased expression of cell cycle-related genes (Supplementary Fig. 2e, f). However, the upregulation of inflammatory cytokines that occurs upon 4-OHT induction is reduced after *MIR31HG* depletion (Fig. 2f). Acosta et al.[12] reported that the transforming growth factor-β (TGF-β) signalling pathway is responsible for the paracrine senescence effect. As expected, RNA levels of TGFβ target genes were upregulated in BJ ER:BRAF upon 4-OHT treatment compared to control proliferating cells (Supplementary Fig. 2g), while *MIR31HG* knock-down did not affect this increase. Moreover, the protein level of pSMAD2, downstream effector of TGFβ signalling, was upregulated during OIS in control as well as in *MIR31HG* knock-down senescent cells (Supplementary Fig. 2h). Similarly, WT BJ cells discretely activated TGFβ signalling when incubated with CM from control senescent cells and also with CM from *MIR31HG* knock-down senescent cells (Supplementary Fig. 2i, j). These results demonstrate that the TGFβ signalling pathway remains active in *MIR31HG* knock-down senescent cells suggesting that its activation could be responsible for the paracrine senescence induction. To determine the role of the SASP in promoting cancer cell invasion, we performed transwell invasion assays using the different CM described above as chemoattractant in the lower chamber of the transwell for MDA-MB-231 breast cancer cells. After 48 h incubation a higher number of MDA-MB-231 cells exposed to CM from senescent cells invaded through the matrigel membrane compared to proliferating CM (Fig. 2g, h). In contrast, the CM of *MIR31HG* knock-down senescent cells did not promote invasion (Fig. 2g, h). These results demonstrate that the paracrine senescence effect is retained in the SASP of *MIR31HG* knock-down senescent cells but not the induction of paracrine invasion. These findings suggest that the effect of *MIR31HG* in the SASP production does not occur at a general level but that only a subset of components is inhibited.

**MIR31HG depletion reduces CEBPB and NF-κB activation in OIS.** As NF-κB and CEBPB regulate the expression of many SASP genes by stimulating their transcription[11,32,33], we investigated the possibility that *MIR31HG* knock-down affected the abundance or the activation of these transcription factors. As expected, the protein levels of CEBPB and activated phosphorylated RELA (p-RELA) (component of the NF-κB complex) increased during BRAF activation (Fig. 3a, b). Interestingly, *MIR31HG* knock-down in senescent cells reduced CEBPB protein expression and activated p-RELA (Fig. 3a, b) without affecting their mRNA levels (Supplementary Fig. 3a). In accordance, binding of CEBPB to the *IL6* promoter, a canonical CEBPB target, decreased significantly upon *MIR31HG* knock-down compare to control senescent conditions (Fig. 3c). Consistently, NF-κB translocation to the nucleus was inhibited upon *MIR31HG* depletion (Fig. 3d and Supplementary Fig. 3b). Moreover, the signalling pathway upstream NF-κB activation was affected in *MIR31HG* knock-down senescent (Supplementary Fig. 3c). These results suggest that *MIR31HG* acts upstream of both CEBPB and NF-κB transcriptional activation.

**MIR31HG depletion decreases IL1A translation in OIS.** IL1A has been reported to function as an upstream regulator of the SASP[34]. In line with this, addition of human recombinant IL1A (hr-IL1A) in our model system induced expression of SASP components at the RNA level (Supplementary Fig. 3d). Blocking IL1A signalling using siRNA against IL1A reduced the levels of CEBPB expression upon senescence as well as the RNA levels of different components of the SASP (Supplementary Fig. 3e, f). To validate the paracrine role of IL1A in the SASP induction, we knocked down IL1A in senescent cells and use this CM to assess

the expression of SASP components by qRT-PCR in recipient BJ and IMR90 cells. The CM from IL1A knock-down senescent cells was not able to induce paracrine expression of SASP components in both cells lines BJ and IMR90 (Supplementary Fig. 3g). Interestingly, while *IL1A* mRNA was not affected by the levels of *MIR31HG* (Fig. 1e), IL1A protein levels were strongly reduced in *MIR31HG* knock-down senescent cells compared to control senescent cells as measured by western blot of the total lysate (Fig. 3e, f) and by immunofluorescence staining (Fig. 3g). ICAM1, which was not affected by MIR31HG depletion at the RNA level (Fig. 1e), did not show any protein decreased, suggesting some degree of specificity (Supplementary Fig. 3h, i).

Addition of human recombinant IL1A (hr-IL1A) rescued the incapacity of *MIR31HG* knock-down senescent cells to induce SASP RNA transcription (Fig. 3h), IL6 secretion (Fig. 3i), and NF-κB nuclear translocation (Supplementary Fig. 3j).

To test whether *MIR31HG* would be regulating IL1A at a translational level, we performed polysome profiling in control senescent cells and senescent cells upon depletion of *MIR31HG*. The profiles obtained upon sucrose gradient separation in both conditions showed no major changes in polysome distribution, indicating no changes in global protein synthesis (Fig. 3j). Analysing the distribution of *IL1A* mRNA through the gradient we observed the majority of the transcript present in the heavy polysome fractions, indicative of high level of translation (Fig. 3k). Remarkably, a significant decrease of *IL1A* mRNA in these fractions was observed in *MIR31HG* knock-down senescent cells compared to control senescent cells (Fig. 3l and Supplementary Fig. 3k). Distribution of *ACTB* mRNA through the gradient did not show any difference (Fig. 3l and Supplementary Fig. 3k). Despite the decrease in mRNA levels of other cytokines such as IL6, the similar distribution of their mRNAs through the gradient indicates an equal translation efficiency suggesting that the effect of *MIR31HG* on translation is specific for *IL1A* (Fig. 3l and Supplementary Fig. 3k). To validate that reduced IL1A protein levels was due to an effect of *MIR31HG* on translation and not other related processes such as protein stability, we performed AHA pulse labelling and coupling of newly synthesized proteins to biotin followed by streptavidin pull-down and immunoblotting. We confirmed that *MIR31HG*-depleted senescent cells contained less newly synthesized IL1A compared to control senescent cells, whereas the amount of GAPDH or Vinculin remained unaltered (Fig. 3m). MTOR signalling has been recently involved in SASP regulation through IL1A signalling[35]. However, *MIR31HG* knock-down did not affect mTOR signalling activation (Supplementary Fig. 3l), indicating an alternative mechanism for *MIR31HG* regulation of IL1A. These findings suggest that *MIR31HG* is implicated in the modulation of the SASP by regulating IL1A translation independent of the mTOR signalling pathway.

**MIR31HG interacts with YBX1.** In order to elucidate the mechanism by which *MIR31HG* exerts its function in regulating the SASP, we purified endogenous *MIR31HG* with its associated proteins from UV-crosslinked senescent BJ ER:BRAF cells using antisense oligonucleotides containing locked nucleic acid (ASOs) complementary to the *MIR31HG* sequence coupled to magnetic beads (Fig. 4a). Oligonucleotides against the RNA sequence of luciferase, not expressed in BJ ER:BRAF cells, were used as control. Mass spectrometry analysis of *MIR31HG*-associated proteins revealed a low number of significantly enriched proteins among which several RNA-binding proteins (Fig. 4b and Supplementary Data 2). *MIR31HG* has been previously shown to interact with IκBα during osteogenic differentiation regulating NF-κB activation[26]. However, we did not detect IκBα in our pulldown experiments. The majority were heterogeneous nuclear

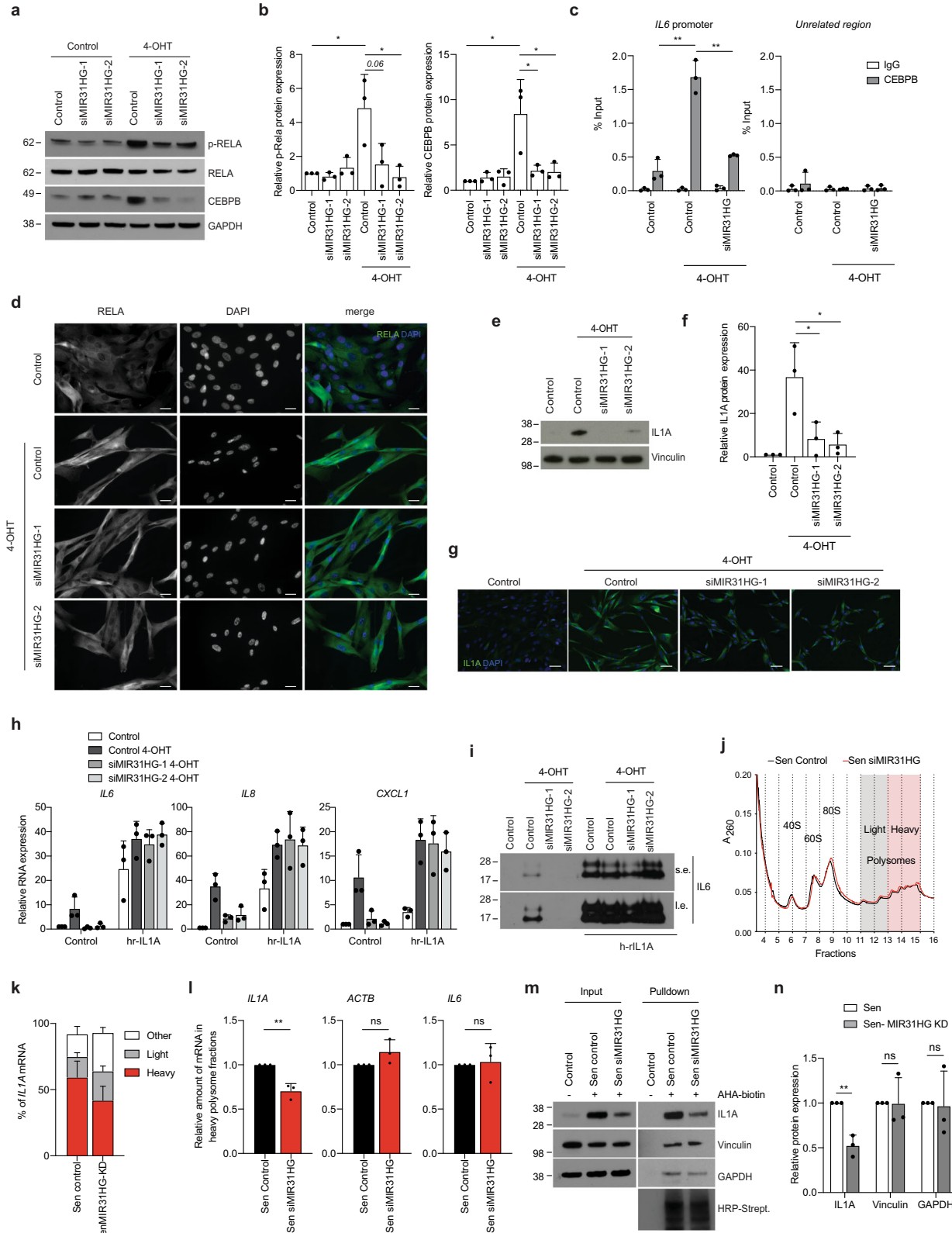

ribonucleoproteins or other predominantly nuclear proteins involved in RNA processing. Recent reports have demonstrated the role of PTBP1 in regulating the SASP composition through the regulation of the alternative splicing of genes implicated in intracellular trafficking[36]. However, since *MIR31HG* localizes mainly in the cytoplasm during OIS (Supplementary Fig 4a) we focused on YBX1 as a protein with defined cytoplasmic functions (Fig. 4b).

**YBX1 knock-down phenocopies *MIR31HG* depletion.** YBX1 has been implicated in several cytoplasmic processes, such as translation or mRNA stability among others[37]. According to previous findings, YBX1 prevents senescence in epidermal progenitors and knock-down of YBX1 induces senescence features in keratinocytes[38]. Depletion of YBX1 in untreated proliferating BJ ER:BRAF resulted in decreased proliferation and p21 increase,

**Fig. 3 *MIR31HG* knock-down decreases IL1A translation. a** Western blot for p-RELA, RELA, CEBPB and GAPDH of BJ ER:BRAF cells (Control or siMIR31HG1-2) treated with ethanol (Control) or 1 μM 4-OHT for 72 h. Molecular weight marker is shown in kDa. **b** Quantification of the p-RELA and CEBPB western blot band intensities relative to GAPDH ($n = 3$). **c** Chromatin immunoprecipitation followed by qPCR for CEBPB binding to the *IL6* promoter and to an unrelated region in BJ ER:BRAF cells (Control or siMIR31HG1) treated with ethanol (Control) or 1 μM 4-OHT for 48 h. The graph shows the percentage of the input that binds CEBPB ($n = 3$). **d** Immunofluorescence analysis for RELA and DAPI of BJ ER:BRAF cells (Control or siMIR31HG1-2) treated with ethanol (Control) or 1 μM 4-OHT for 72 h. Representative images are shown in the figure ($n = 3$). Scale bar: 20 μm. **e** Western blot for IL1A in BJ ER:BRAF cells (Control or siMIR31HG1-2) were treated with ethanol (Control) or 1 μM 4-OHT for 72 h. Molecular weight marker is shown in kDa ($n = 3$). **f** Quantification of IL1A western blot band intensities relative to Vinculin ($n = 3$). **g** Immunofluorescence of IL1A and DAPI in BJ ER:BRAF cells (Control or siMIR31HG1-2) treated with ethanol (Control) or 1 μM 4-OHT for 72 h. Representative merged images are shown in the figure ($n = 3$). Scale bar: 50 μm. **h** Relative RNA expression for the indicated SASP factors in BJ ER:BRAF cells transfected with the indicated siRNAs (Control or siMIR31HG1-2), treated with ethanol (Control) or 1 μM 4-OHT for 48 h, in the absence or presence of 10 ng/ml of h-rIL1A for 2 h before RNA extraction ($n = 3$). **i** Western blot for IL6 from precipitated protein from the media in the conditions indicated in **h** in the absence or presence of 10 ng/ml of h-rIL1A for 24 h before harvesting the CM ($n = 3$). S.e. short exposure, l.e. long exposure. **j** Polysome profile performed by sucrose gradient separation in senescent BJ ER:BRAF control cells (Sen Control, black) or siMIR31HG cells (Sen siMIR31HG, red) treated with 1 μM 4-OHT for 72 h. Y-axis shows the absorbance at 260 nM and X-axis shows the number of the fractions collected. Red area shows the heavy polysome fractions and grey area shows the light polysome fractions. One representative experiment is shown in the figure ($n = 3$). **k** Distribution of the amount of *IL1A* mRNA in the heavy polysome (red) light polysome (grey) or other fractions (white) from the experiment in **g**. **l** Relative amount of *IL1A, IL6* and *ACTB* mRNA present in the heavy polysome fractions in the experiment described in **j**. **m** Western blot analysis of newly synthesized IL1A, Vinculin and GAPDH levels in control senescent cells and *MIR31HG*- depleted senescent cells purified by AHA pulse-labelling and coupling to biotin followed by streptavidin pull-down. As control, proliferating cells without AHA labelling is shown. HRP-streptavidin shows the uniform labelling of newly synthesized proteins coupled to biotin. A representative experiment is shown ($n = 3$). **n** Quantification of IL1A, Vinculin and GAPDH western blot band intensities from pulldown samples in senescence (Sen, white) and senescence MIR31HG KD cells (Sen-MIR31HG KD, grey) ($n = 3$). All statistical significances were calculated using two-tailed Student's *t*-tests, \*$p < 0.05$; \*\*$p < 0.01$; ns non-significant. All error bars represent means ± s.d. Source data are provided as a Source Data file.

although p53 levels were strongly downregulated (Supplementary Fig. 4b, c). Interestingly, knock-down of YBX1 in BRAF-induced senescent cells mimicked *MIR31HG* knock-down phenotype. SASP components were decreased at RNA level in YBX1 knock-down senescent cells compared to control senescent cells (Fig. 4c, d and Supplementary Data 3). Likewise, reduction in the secretion of IL6 and CXCL1 upon YBX1 knock-down were comparable to the levels following *MIR31HG* knock-down (Figs. 1e and 4e). Furthermore, IL1A protein levels were strongly reduced (Fig. 4f, g) whereas the mRNA level remained unaltered (Fig. 4d).

Formaldehyde crosslinked RNA immunoprecipitation (CLIP) of GFP-tagged YBX1 validated its interaction with *MIR31HG* (Fig. 4h). As a positive control, we found YBX1 binding its own RNA as previously reported[39]. Interestingly, YBX1 binds *IL1A* mRNA whereas other cytokine mRNAs are bound to a lesser extent (Fig. 4h). Furthermore, no binding to abundant nuclear RNAs (*MALAT1*) or the mitochondrial RNAs previously used as negative controls[40] was identified. To identify the YBX1-binding sites in *MIR31HG* we performed in silico prediction based on previous iCLIP studies[41,42], and identified five putative binding sites (Fig. 4i). We generated *MIR31HG* mutants harbouring deletions including the predicted binding sites (Fig. 4i) to perform electrophoretic mobility shift assays (EMSA), incubating the corresponding in vitro-transcribed RNA with recombinant YBX1. *ND4*, a mitochondrial transcript previously used as a negative control (Fig. 4h), did not produce a shift indicating the absence of binding, whereas the full-length *MIR31HG* (WT) resulted in a band shift when incubated with increasing amounts of YBX1 (Fig. 4j). Mutant 1 and 5 (Mut1 and Mut5) were able to bind YBX1 to the same extent as the WT, whereas mutants 2, 3 and 4 (Mut2, Mut3 and Mut4) were not (Fig. 4j). To validate these results, we performed competition assays incubating $^{32}$P-labelled *MIR31HG* WT transcript with increasing amounts of unlabelled Mut5 or Mut3. We observed that only mut5 competed for the binding of YBX1 (Fig. 4k). These results indicate that the predicted binding site BS3 in *MIR31HG* is required for YBX1 binding.

To study the biological function of the *MIR31HG* and YBX1 interaction, we analysed the stability and localization of *MIR31HG* upon YBX1 knock-down, since stabilization of RNAs is a well-described function of cytoplasmic YBX1 (refs. [43,44]). We

did not observe changes in *MIR31HG* expression nor localization upon YBX1 knock-down (Supplementary Fig. 4d, e). Moreover, protein levels of YBX1 did not change upon depletion of *MIR31HG* (Supplementary Fig 4f).

**YBX1 is phosphorylated at serine 102 by RSK in OIS.** Several modifications which can affect YBX1 function have been reported[45]. Phosphorylation of serine in position 102 has been shown to be involved in the regulation of translation[46]. We therefore studied the phosphorylation status of YBX1 and the putative implication of *MIR31HG* in YBX1 regulation. In order to analyse p-YBX1 in BRAF-induced senescence, we used an antibody that recognizes p-YBX1 at position S102 (p-YBX1$^{S102}$). Interestingly, we detected increased levels of this modification at different time points after BRAF induction (Fig. 5a and Supplementary Fig. 5a). Different kinases have been reported to be responsible for S102 YBX1 phosphorylation such as AKT and RSK[47,48]. We analysed the levels of activated AKT at different time points after 4-OHT induction and we observed that p-AKT decreased over time to nearly undetected levels already after 24 h treatment (Supplementary Fig. 5b). The kinase RSK, however, is activated at early points during senescence induction and remain active at later time points (Fig. 5a and Supplementary Fig. 5a) suggesting that YBX1 might be a substrate for RSK during OIS. To investigate the role of RSK in YBX1 phosphorylation in BRAF-OIS, we treated the cells with specific RSK inhibitors. Treatment of senescent cells with the inhibitor FMK reduced the levels of p-YBX1$^{S102}$ (Fig. 5b and Supplementary Fig. 5c). Moreover, BI-D780, a more specific RSK inhibitor, was able to completely abolish phosphorylation of YBX1 in senescent cells (Fig. 5b and Supplementary Fig. 5c). Furthermore, overexpression of RSK resulted in an increased level of p-YBX1$^{S102}$ (Supplementary Fig. 5d). Our results conclude that YBX1 is phosphorylated by RSK kinase in a BRAF-dependent manner.

**pYBX1$^{S102}$ induces translation of IL1A.** It has been previously shown that YBX1 can promote translation of specific transcripts[49,50]. We next wondered whether p-YBX1$^{S102}$ could impact *IL1A* translation. As expected, BRAF-induced senescence

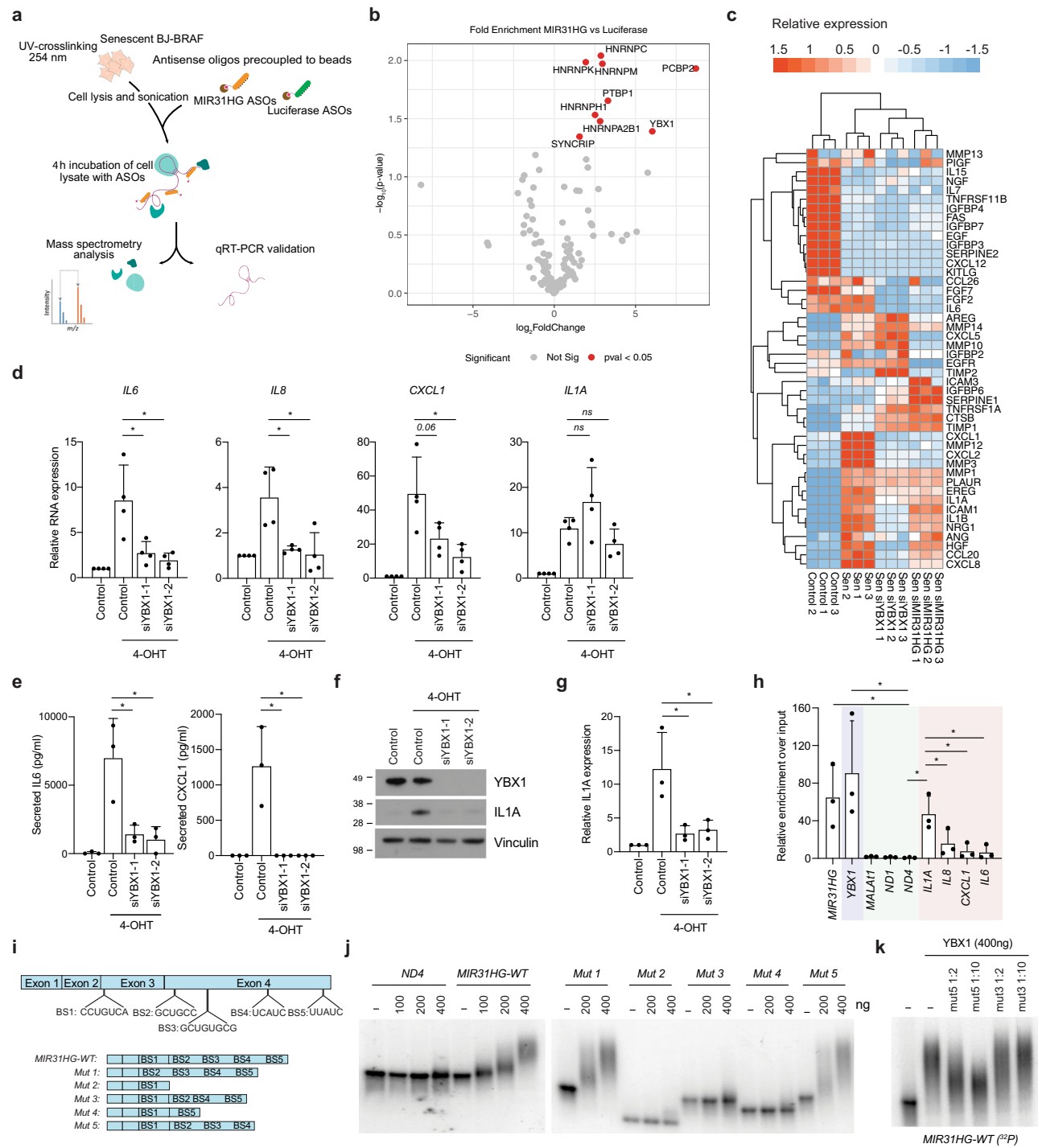

resulted in an upregulation of IL1A protein levels compared to control untreated cells (Fig. 5c and Supplementary Fig. 5e). Interestingly, treatment with the RSK inhibitor BI-D1780 failed to upregulate IL1A during senescence (Fig. 5c and Supplementary Fig. 5e). YBX1 has been recently implicated in cytokine translation by binding the 3′UTR of their mRNAs[38]. Using reporter plasmids harbouring the 3′UTR of IL1A after luciferase gene we demonstrated that a WT version of YBX1 is able to increase luciferase translation in response to 4-OHT induction (Fig. 5d). A mutant that is not able to be phosphorylated at S102 (S102A) failed to induce translation whereas a phosphomimic mutant (S102D) increased translation even in the absence of 4-OHT (Fig. 5d and Supplementary Fig. 5f). These results suggest that p-

YBX1 binds *IL1A* 3′UTR to increase its translation. We assessed the binding capacity of the different phosphorylated versions of YBX1 to *IL1A* mRNA and to *MIR31HG* by RIP. All the proteins showed a similar binding to their targets suggesting that phosphorylation at S102 is not affecting YBX1 binding capacity (Fig. 5e and Supplementary Fig. 5g). We next characterized the role of the p-YBX1$^{S102}$ in invasion using transwell assays. The CM from ethanol treated (control) or 1 μM 4-OHT BJ ER:BRAF cells transfected with control siRNAs or siRNAs against RSK, YBX1 or MIR31HG were used as chemoattractant in the lower chamber of the transwell for MDA-MB-231 breast cancer cell invasion. After 24 h a higher number of MDA-MB-231 cells exposed to CM from senescent cells invaded through the matrigel

**Fig. 4 *MIR31HG* interacts with YBX1 and YBX1 knock-down phenocopies *MIR31HG* depletion. a** Schematic representation of the pulldown procedure. **b** Cellular extracts from BJ ER:BRAF (Control or siMIR31HG) treated with 1 μM 4-OHT for 72 h were incubated with antisense oligos, enriched proteins extracted and subjected to LC-MS analysis (see 'Methods'). The volcano plot highlights proteins enriched in the *MIR31HG* pulldown analysis compared to a luciferase control pull down. Marked in red are protein with $p$ value <0.05. **c** Heat map showing the relative expression of significant differentially expressed SASP genes in BJ ER:BRAF cells (control or siMIR31HG) treated with ethanol (Control 1-3) or 1 μM 4-OHT for 48 h (Sen 1-3, Sen siMIR31HG 1-3, Sen YBX1-KD1-3). **d** Relative RNA expression of selected components of the SASP normalized to housekeeping genes (*HPRT1* and *RPLP0*) in BJ ER:BRAF cells transfected with the indicated siRNAs (Control or siYBX1, 1–2), treated with ethanol (Control) or 1 μM 4-OHT for 48 h. The graph shows results compared to control ethanol-treated set to 1 ($n = 4$). **e** Secreted IL6 and CXCL1 (pg/ml) measured by ELISA in the CM of BJ ER:BRAF (Control or siYBX1, 1–2) treated for 72 h with ethanol (Control) or 1 μM 4-OHT ($n = 3$). **f** Western blot for IL1A in BJ ER:BRAF cells (Control or siYBX1, 1–2) were treated with ethanol (Control) or 1 μM 4-OHT for 72 h. Molecular weight marker is shown in kDa ($n = 3$). **g** Quantification of the IL1A western blot band intensities relative to Vinculin ($n = 3$). **h** Relative RNA binding using GFP-YBX1 in 4-OHT-treated BJ ER:BRAF, represented as the percentage of the input bound relative to an empty GFP cell line. *YBX1* was used as a positive control (blue), *MALAT1*, *ND1*, and *ND4* binding as negative controls (green) and cytokines binding (pink). The results are shown as the percentage of input relative to an empty GFP cell line treated in the same conditions ($n = 3$). **i** *Top*, schematic representation of MIR31HG with the putative YBX1-binding sites (BS) found in published iCLIP data. BS1-3 were found in Goodarzi et al. iCLIP data and BS5-4 were retrieved from Wu et al. dataset. *Bottom*, representation of *MIR31HG* truncations (Mut1 to Mut5). **j** EMSA showing the in vitro binding of 2 nM of the corresponding radiolabelled transcript and the indicated amounts of recombinant YBX1. BSA at the highest concentration (400 ng) was used as a negative control ($n = 3$). **k** EMSA showing the in vitro binding of 400 ng of YBX1 to 2 nM of the radiolabelled *MIR31HG* wild type in competition with the indicated amounts of unlabelled Mut5 or Mut3 ($n = 3$). All statistical significances were calculated using two-tailed Student's t-tests, *$p < 0.05$; ns non-significant. All error bars represent means ± s.d. Source data are provided as a Source Data file.

membrane compared to control CM (Supplementary Fig. 5h, i). In contrast, the CM of RSK and YBX1 knock-down senescent cells promoted limited invasion as well as the CM of *MIR31HG*-depleted cells (Supplementary Fig. 5h, i). These results show that RSK and YBX1 are required for invasion. In order to further analyse the role of YBX1 phosphorylation, we performed trans-well assays using BJ ER:BRAF cells expressing the different YBX1 constructs described above. Overexpression of all forms of YBX1 increased invasion independently of its phosphorylation status (Fig. 5f, g). However, under senescence conditions, over-expression of WT YBX1 promoted invasion to a similar extent as that of the S102D phosphor-mimic mutant, whereas the invasion was reduced when overexpressing the S102A mutant (Fig. 5f, g). Although endogenous YBX1 was depleted using siRNAs targeting the 3′UTR, residual YBX1 might explain the fact that all the constructs show increased invasion in 4-OHT-treated conditions compared to control. Altogether these results suggest a role of p-YBX1 in promoting invasion.

**MIR31HG knock-down inhibits YBX1–RSK interaction**. We next examined whether *MIR31HG* might have an impact on YBX1 phosphorylation. We performed cellular fractionation to analyse the localization of phosphorylated YBX1 in senescent cells and in senescent cells where *MIR31HG* was depleted. Interestingly, we observed that upon *MIR31HG* knock-down p-YBX1 is reduced in the cytoplasm (Fig. 6a, b), suggesting a role for *MIR31HG* in the phosphorylation process of YBX1. Several cytoplasmic lncRNAs have been shown to act as scaffolds for bringing molecules into close proximity. In order to analyse whether *MIR31HG* was binding both YBX1 and its kinase RSK, we performed native RIP using GFP-tagged version of RSK. We could not detect RSK binding to *MIR31HG* (Supplementary Fig. 6a, b). To further analyse the role of *MIR31HG* in the interaction of YBX1 with its kinase RSK, we performed proximity ligation assays (PLAs) using antibodies against total YBX1 and total RSK. Importantly, the interaction was higher in senescence compared to control (Fig. 6c, d). Interestingly, *MIR31HG* knock-down reduced the level of interaction confirming our hypothesis that *MIR31HG* mediates YBX1 interaction with its kinase RSK during OIS (Fig. 6c, d). In order to corroborate this result, we have performed in vitro kinase assays using recombinant RSK and recombinant YBX1, and observed that YBX1 is phosphory-lated in a RSK-dependent manner (Supplementary Fig. 6c). We could detect RSK auto-phosphorylation as previously shown[51]

(Fig. 6e and Supplementary Fig. 6c). Addition of in vitro-transcribed *MIR31HG* resulted in a moderate but significant increase in p-YBX1 phosphorylation compared to *ND4* and *Mut3*, lacking YBX1-binding site (Fig. 6e). To demonstrate that the interaction between YBX1 and *MIR31HG* is responsible for YBX phosphorylation, we created BJ ER:BRAF cell lines expressing doxycycline-inducible *MIR31HG* constructs WT, Mut5 and Mut3 (Supplementary Fig. 6d). Interestingly, overexpression of WT and Mut5 transcripts showed increased levels of p-YBX1 compared to an empty cell line and Mut3 overexpression, which was unable to bind *MIR31HG* (Figs. 4k and 6f, g). These results confirm that the interaction between *MIR31HG* and YBX1 facilitates its phosphorylation.

## Discussion

The SASP is a hallmark of senescent cells and responsible for mediating the patho-physiological effects in the surrounding tis-sues. For a long time, efforts in the field have been focused on trying to induce senescence in cancer cells in order to prevent cancer progression[7,52,53]. In contrast hereto, more recent reports have shown that targeting senescent cells strongly improve age-related diseases[20,54,55]. Although recent work has extended our knowledge of the signalling network upstream the transcriptional induction of the SASP components[56–60], the complexity and diversity of the SASP suggest that additional regulators might be involved. It is crucial to acquire further knowledge of the detailed mechanism to be able to design therapies for cancer, inflammation and aging treatment. Here, we contribute to understanding the SASP regulation by describing the molecular mechanism by which the lncRNA *MIR31HG* regulates a subset of the SASP components during BRAF-induced senescence by modulating *IL1A* translation. It is known that most of the SASP components are regulated at the transcriptional level, although a small fraction may additionally be regulated by post-transcriptional mechanisms[35,57]. In this study, we focus on the senescence process initiated following the expression of a mutated *BRAF*. This is of importance as somatic mutations in *BRAF* occur in approximately 7% of human cancer[61]. We observed that *MIR31HG* expression is higher in thyroid and colorectal cancer tumour samples harbouring *BRAF* mutations, as compared to *BRAF* wild type, suggesting the rele-vance of this lncRNA in cancer.

We find that the secretome from BJ ER:BRAF senescent cells and BJ ER:BRAF *MIR31HG* knock-down senescent cells show differences in the levels of key SASP components. Whereas most

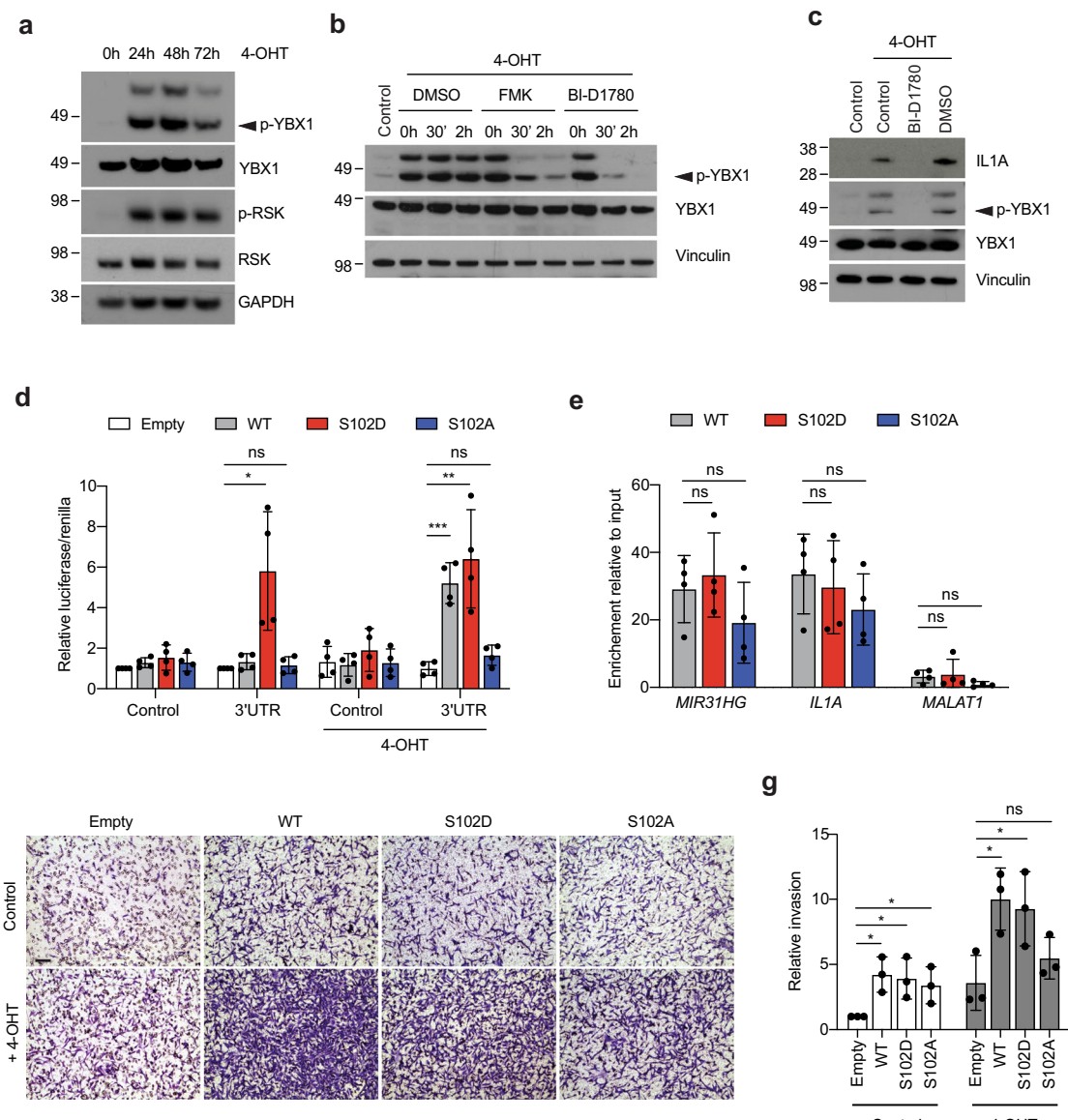

**Fig. 5 Phosphorylated YBX1 induces IL1A translation. a** Western blot for p-RSK1, RSK1, p-YBX1, total YBX1 and Vinculin in BJ ER:BRAF cells treated with 1 µM 4-OHT for the indicated time. Molecular weight marker is shown in kDa ($n = 3$). **b** Western blot for p-YBX1, total YBX1 and Vinculin in BJ ER:BRAF cells untreated (control) or treated with 1 µM 4-OHT and the indicated inhibitor at 1 µM for the indicated time. DMSO is used as vehicle control. Molecular weight marker is shown in kDa ($n = 3$). **c** Western blot for p-YBX1, total YBX1, IL1A and Vinculin in BJ ER:BRAF cells untreated (control) or treated with 1 µM 4-OHT and BI-D1780 at 1 µM for 72 h. Molecular weight marker is shown in kDa ($n = 4$). **d** Relative luciferase measurement in BJ ER:BRAF cells expressing a doxycycline-induced empty, wild type (WT), mutant S102A or mutant S102D versions of YBX1 were transfected with a siRNA against YBX1 and reporter constructs pGL3-promoter (control) or pGL3-promoter-3'UTR (3'UTR) containing the 3'UTR of IL1A mRNA and treated with ethanol (control) or with 1 µM 4-OHT for 48 h. The graph shows the Luciferase values normalized to renilla relative to the empty cell line in the absence of 4-OHT set as 1 ($n = 4$). **e** RIP analysis using a GFP-tagged version of WT YBX1, S102A mutant or S102D mutant in formaldehyde crosslinked cells induced with doxycycline and treated with 1 µM 4-OHT for 72 h. The graph shows YBX1 binding to *MIR31HG*, *IL1A* and *MALAT* as a negative control. The results are shown as the percentage of input relative to an empty GFP cell line treated in the same conditions ($n = 4$). **f** MDA-MB-231 invading cells through a matrigel membrane in contact with the CM from BJ ER:BRAF cells expressing a doxycycline-induced empty, wild type, mutant S102A or mutant S102D treated for 72 h with ethanol (Control) or 1 µM 4-OHT. Representative images are shown in the figure. Scale bar: 50 µm. **g** Quantification of the invading cells from (**f**) relative to control ethanol-treated cells ($n = 3$). All statistical significances were calculated using two-tailed Student t-tests, *$p < 0.05$; **$p < 0.01$; ***$P < 0.001$; ns non-significant. All error bars represent means ± s.d. Source data are provided as a Source Data file.

of the genes deregulated at the transcription level are also altered at the protein level, a small proportion is regulated only at RNA or protein level. This, together with the comparable polysome distribution profiles of these cell lines, suggests no changes in global translation upon *MIR31HG* depletion. Interestingly, IL1A, an upstream regulator of the SASP, is reduced at the protein level following *MIR31HG* knock-down whereas the mRNA level is not

significantly altered. No IL1A was detected in the conditioned media by mass spectrometry or by ELISA, likely due to the fact that IL1A remains attached to the membrane during OIS as previously described[34]. In our cellular system addition of hr-IL1A induces the transcription of several cytokines and instigates the SASP. Intriguingly, it also rescues the decreased transcription of SASP components caused by *MIR31HG* depletion during BRAF-

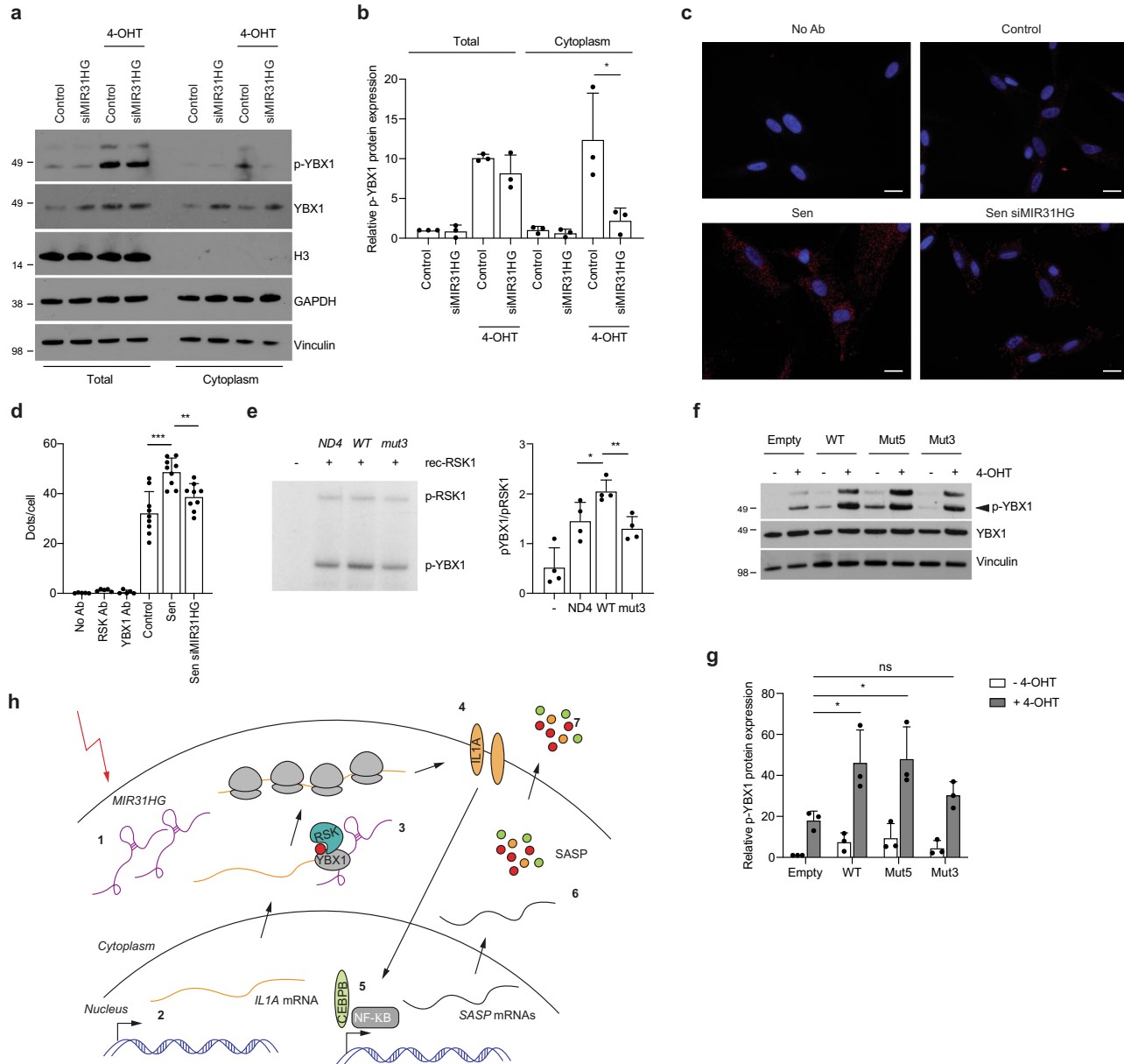

**Fig. 6 *MIR31HG* promotes YBX1 phosphorylation by facilitating YBX1 interaction with its kinase RSK. a** Western blot analysis for p-YBX1, total YBX1, GAPDH and H3 in total and cytoplasmic fractions of BJ ER:BRAF cells (Control or siMIR31HG1-2) treated with ethanol (Control) or 1 μM 4-OHT for 72 h (n = 3). **b** Quantification of the pYBX1 band intensities relative to total YBX1 and Vinculin from three independent experiments. **c** PLA representative images showing the interaction (red dots) between YBX1 and RSK in BJ ER:BRAF cells (Control or siMIR31HG1) were treated with ethanol (Control) or 1 μM 4-OHT for 72 h. DAPI staining (blue) shows the nucleus. **d** Quantification of the number of interactions (dots) per cell in the conditions indicated in a representative experiment from four independent replicates (n = 4). Scale bar: 20 μm. **e** *Left*, in vitro kinase experiment incubating recombinant YBX1 in the presence of $^{32}$P-ATP and recombinant RSK1 in the presence of the indicated in vitro-transcribed RNA, run in 4–12% NuPAGE Bis–Tris gel and exposed to an Amersham Hyperfilm ECL film. *Right*, quantification of the band intensities represented as p-YBX1 related to p-RSK (n = 4). **f** Western blot for p-YBX1, total YBX1 and Vinculin in BJ ER:BRAF cells expressing a doxycycline-induced empty, *MIR31HG* WT, Mut3 or Mut5 treated for 36 h with ethanol (Control) or 1 μM 4-OHT. **g** Quantification of the p-YBX1 band intensities relative to total YBX1 and Vinculin (n = 3). **h** Scheme of the working model: (1) During OIS *MIR31HG* is induced and locates in the cytoplasm. (2) At the same time, transcription of *IL1A* mRNA is induced. (3) *IL1A* mRNA is exported to the cytoplasm where it interacts with YBX1. *MIR31HG* facilitates YBX1 interaction with the kinase RSK promoting its phosphorylation which in turns promotes *IL1A* translation. (4) IL1A signalling induces the transcription of other SASP mRNAs through CEBPB and NF-κB. (5) These RNAs will be translated in the cytoplasm and the SASP components will be secreted (6). Statistical significance was calculated using two-tailed Student's t-tests, *p < 0.05; **p < 0.01; ***p < 0.001. Error bars represent means ± s.d. Source data are provided as a Source Data file.

induced senescence. These results strongly support the idea of *MIR31HG* regulating IL1A as an upstream regulator of the SASP transcriptional program. Work from the Campisi lab has demonstrated that the translation of *IL1A* during RAS-induced senescence is regulated by mTOR[35]. Although the mTOR

pathway is increased in our OIS model, it is not altered in *MIR31HG* knock-down conditions. Instead, we found the lncRNA *MIR31HG* to affect *IL1A* translation through YBX1. Endogenous *MIR31HG* pull down and subsequent validations by RIP and EMSA confirm that YBX1 is a *MIR31HG*-binding

partner. Importantly, YBX1 depletion mimics the *MIR31HG* knock-down phenotype in BRAF-induced senescence. This includes the reduced translation of *IL1A* and therefore the downregulation of downstream SASP components at RNA level, such as CXCL1, IL8 or IL6 among others. Recent work has shown that YBX1 prevents cytokine translation in cycling keratinocytes by binding their 3′UTR. In these studies, YBX1 knock-down results in increased translation of several cytokines and therefore induction of senescence[38]. Knock-down of YBX1 in cells subjected to BRAF-induced senescence resulted in reduced RNA levels of most of the cytokines previously annotated as SASP components[13]. Interestingly, in BRAF-induced senescence we observe that YBX1 binds a high fraction of the IL1A mRNA while binding other cytokines to a lesser extent. YBX1 binds AU-rich motifs present in the 3′UTR of cytokine mRNAs[38,62], which is consistent with our *IL1A* 3′ UTR luciferase reporter assays. However, the stronger binding of YBX1 to *IL1A* mRNA compared to other cytokines show some degree of specificity towards *IL1A* mRNA. The mechanism underlying this specificity remains unknown. In a recent study, Dominguez et al.[63] determined the importance of contextual features in RNA recognition by RBPs. Perhaps RNA secondary structures, the flanking nucleotide composition or the proximity to other RNA-binding proteins might be influencing the binding of YBX1 to the *IL1A* mRNA.

We observe an increase in p-YBX1$^{S102}$ in BRAF-induced senescence and, importantly, p-YBX1$^{S102}$ is reduced in the cytoplasm upon *MIR31HG* knock-down. Our results indicate that this post-transcriptional modification of YBX1 is involved in promoting *IL1A* translation during BRAF-induced senescence through its 3′UTR, which could explain the previously reported opposite roles of YBX1 in cytokine translation[38]. The role of p-YBX1 in translation remains to be fully understood. It is known that YBX1 binds the mRNA cap structure by displacing the eukaryotic translation factor 4E (eIF4E) and eIF4G and hence promotes translational repression[44,64]. p-YBX1$^{S102}$, on the other hand, binds the mRNA cap structure less tightly allowing translation of repressed mRNAs[46]. However, not much is known about its role in promoting translation. Our luciferase reporter assay indicates that during senescence, p-YBX1$^{102}$ may have a role in promoting *IL1A* translation through the 3′UTR since the phosphomutant YBX1$^{S102A}$ fails to induce luciferase induction. Importantly, invasion is reduced when overexpressing YBX1$^{S102A}$ as compared to p-YBX1$^{102}$ and YBX1$^{S102D}$. Although we have observed increased invasion upon YBX1 overexpression independently on its phosphorylation status, confirming previous findings of YBX1 involved in metastasis and invasion[49,65,66], this result suggests that the YBX1 reduced phosphorylation is, at least, partly responsible for the SASP-induced invasion observed in *MIR31HG*-depleted senescent cells. Mutation of S102 did not alter YBX1 binding capability to *IL1A* or other mRNAs, consistently with previous reports demonstrating that phosphorylation at S102 does not affect the general RNA binding capacity of YBX1 (ref. [44]). Phosphorylation of translation factors are determining events occurring during translation regulation[67] and perhaps p-YBX1 is involved in differentially recruiting other translation factors with a role in *IL1A* mRNA translational regulation. Further phospho-proteomic studies would help to address this question. We cannot exclude that phosphorylation at other residues occur during senescence. In fact, S165 and S176 have been shown to activate NF-κB signalling[68,69]. It would be interesting to investigate whether these modifications have any impact in senescence or in translation regulation.

LncRNAs can modulate post-transcriptional modifications of proteins[70–73]. Our results suggest that during BRAF-induce senescence RSK appears to be the main kinase responsible for YBX1 phosphorylation. PLA experiments show an increase in RSK–YBX1 interaction during OIS correlating with an increased level of p-YBX1. Despite at a lower level, the presence of interaction in proliferating conditions suggests that the interaction may occur without BRAF induction. The signalling cascade activated during OIS might be necessary to phosphorylate YBX1 at least at S102. We cannot exclude that RSK phosphorylates other residues in proliferating cells. Nevertheless, our results indicate that the interaction between *MIR3HG* and YBX1 is important for the phosphorylation of YBX1 by RSK and its role in senescence, when the lncRNA is expressed and predominantly located in the cytoplasm[28].

YBX1 is a very abundant protein whereas *MIR31HG* is expressed at low levels. The stoichiometry does not correlate with a static model in which *MIR31HG* acts as a platform for YBX1. Moreover, we do not detect *MIR31HG* directly binding RSK, likely excluding a role for *MIR31HG* as scaffold bridging together YBX1 and its kinase. Our results suggest that *MIR31HG*-YBX1 is a dynamic interaction that might infer an optimal conformation of YBX1 to facilitate its phosphorylation. Although the in vitro kinase experiments support this model, structural studies or more exhaustive phospho-proteomic analysis would be required to confirm this hypothesis.

The SASP has been reported to either limit or promote tumour progression. Dissociation of the good and bad sides of the SASPs is difficult since many of its components can have both roles depending on the cellular context[14]. Besides, many of the senescence effectors are activated by the same pathways that activate the SASP. Therefore, the identification of factors that can affect the SASP without altering the tumour-suppressive effects associated with senescence is a promising strategy for senescence-related therapies. Several reports have described factors that can uncouple the senescence growth arrest from the SASP[35,36,57,59]. Here, we identify an lncRNA that, when located in the cytoplasm during OIS, is responsible for the production of a distinct subset of SASP components. Its depletion during OIS leads to a decrease of interleukins, chemokines and other factors preventing invasion in vitro without reverting growth arrest. Interestingly, we have previously described a mechanism for the nuclear *MIR31HG* in proliferating cells in repressing p16$^{INK4A}$ expression by recruiting polycomb group proteins[28]. Altogether, our results suggest that the lncRNA *MIR31HG* has a dual role in inducing and buffering senescence depending on the cellular localization, which highlights the complexity of the regulation of senescence and the SASP. Interestingly, *MIR31HG* has been reported to be upregulated in different types of cancer[74–76]. The fact that depletion of *MIR31HG* induces p16 expression without activation of a SASP response[28] and decreases pro-invasive factors during SASP induction makes it an interesting target for therapeutic purposes. Inhibition of *MIR31HG* as well as other anti-SASP therapies could be potentially used to ameliorate the detrimental effects of senescence cells.

## Methods

**Cell cultures, treatments, siRNA transfections**. The human diploid cell lines TIG3 and BJ were immortalized using pbabe-hTERT and pMSCV-ER:B-RAF retroviral constructs. Hek-293T were transfected using calcium phosphate transfection with 10 μg of each plasmid. Forty-eight hours later, virus were harvested and filtered through a 0.45-μm pore filter. Confluent BJ and TIG3 cells were transduced using 1/3 of the viral supernatant and 8 μg/μl polybrene. Twenty-four hours post-transduction the cells were selected with 50 μg/ml neomycin (h-TERT) and 5 μg/ml blasticidine (B-RAF).

BJ ER:BRAF, TIG3:BRAF, BJ wild type (WT), IMR90 and MDA-MB-231 cells were maintained in Dulbecco's modified Eagle's medium (Invitrogen) supplemented with 10% foetal bovine serum (FBS) (Hyclone) and penicillin/streptomycin (Invitrogen).

BJs and TIG3 were authenticated using IDEXX services (IMPACT II PCR Profile; CellCheck 9—human (9 Marker STR Profile and Inter-species Contamination Test). MDA-MD-231 were obtained from Janine Erler (BRIC,

University of Copenhagen) and IMR90 from Maite Huarte (CIMA, Pamplona, Spain). Senescence was induced by treatment with 1 μM 4-hydroxytamoxifen (4-OHT, Sigma) 24 h after cell seeding for at least 48 h (unless otherwise indicated). siRNA oligonucleotides were transfected at a final concentration of 50 nM by reverse transfection using RNAiMAX (Invitrogen) according to the manufacturer's instructions for 48 or 72 h. The siRNAs sequences are listed in Supplementary Table 1. In case of senescence induction, 1 μM 4-OHT was added 24 h after transfection to fresh media for 48 h for RNA analysis or 72 h for protein analysis. RSK inhibitors, BI-D1780 (Axon 1528) and FMK (Gift from Morten Frodin), were used at 10 μM for the time indicated in each experiment.

*Doxycicline-inducible cell lines. YBX1-GFP and RSK-GFP overexpressing cell lines:* To generate lentiviral constructs, we cloned the coding sequence of YBX1 amplified from cDNA, into the inducible vector pLVX-TetOne Puro Vector (Clontech) that already contained GFP-fusion protein. We use In-Fusion HD Cloning designing primers according to the manufacturer's indications (see primer sequences in Supplementary Table 1).

For the lentiviral production, HEK293T cells were transfected using lipofectamine 2000 (Lifetechnologies) with 7 μg of pLVX-YBX1-GFP or pLVX-empty-GFP, together with 6 μg of VsVg and 5 μg Pax8 viral plasmids. After 48 h supernatants containing the viral particles were filter through a 0.45 μm filter. Confluent BJ ER:BRAF cells were then transduced with 1/3 of the viral supernatant and 8 μg/ml polybrene. Twenty-four hours post-transduction the cells were selected with 1 μg/ml puromycin. Cells were maintained in tetracycline-free tested serum (Clontech). For the expression of YBX1, 100 ng/ml doxycycline was used for 72 h unless otherwise indicated. To generate the phosphomutant (S102A) and phosphomimic (S102D) versions of the protein, directed mutagenesis was performed using QuikChange II XL Site-Directed Mutagenesis Kit (Agilent) directly from the pLVX-YBX1-GFP construct. Primers with the respective mutations were designed according to the manufacturer's indications (see sequences in Supplementary Table 1). The pLVX-RSK-GFP cell line was generated as described for YBX1, amplifying the coding sequence from cDNA with specific primers (sequences in Supplementary Table 1).

*MIR31HG overexpressing cell lines:* To generate the *MIR31HG*-overexpressing cell lines we cloned the full-length *MIR31HG* (NR_027054.2), the Mutant 3 or the Mutant 5 sequences into the inducible vector pLVX-TetOne Puro Vector (Clontech) as described above. Primers are shown in Table 1.

**MIR31HG expression analysis in tumour samples.** Gene expression (RNAseq) and somatic mutation data for tumours in The Cancer Genome Atlas (TCGA Pan-Cancer) were downloaded from UCSC Xena platform[77]. Thyroid carcinoma (THCA) and colorectal adenocarcinoma (COAD & READ) were used for analysis because these tumours have high frequency of BRAF mutations (>10%) and also express MIR31HG (CRC BRAF_wt = 309; CRC BRAF_mutant = 49; THCA BRAF_mutant = 290; THCA BRAF_wt = 199). Expression values for all the samples were converted to log2(fpkm + 1) unit. Wilcoxon test was performed to compare the expression differences in the BRAF mutant and BRAF wild-type tumours, using R.

**RNA extraction and qRT-PCR analysis.** Total RNA was isolated using Trizol reagent (Invitrogen), treated with TURBO DNase (Ambion, Lifetechnologies) and reverse transcribed using TaqMan Reverse Transcription kit (Applied Biosystems) with random hexamer primers. Quantitative real-time PCRs were performed using Syber Green PCR Fast PCR Master Mix 2x and Step One Plus real-time PCR and software v2.3 (Applied Biosystems). The housekeeping genes HPRT1 and RPLP0 were used for normalization of qRT-PCR data, unless otherwise stated.

To calculate the number of molecules per cell, *MIR31HG* was in vitro transcribed (see In vitro transcription and labelling). The concentration of the transcript was measured by $A_{260}$ values and converted to the number of copies using the molecular weight of the RNA. Dilutions of this transcript were retro-transcribed as described above and the complementary DNA was used as standard curve for qPCR. The primers used for qPCR are listed in Supplementary Table 1.

**Western blot analysis.** Cells were seeded and reverse transfected in six-well plates (NUNC). In case of senescence, cells were treated the following day with 1 μM 4-OHT. After 72 h cells were harvested, washed once with phosphate-buffered saline (PBS) and the pellets lysed in RIPA buffer (150 nM NaCl, 0.1% sodium deoxycholate, 0.1% sodium dodecyl sulfate, 50 mm Tris-HCl (pH 8), 1 mM EDTA) containing protease inhibitors (Complete Mini Protease Inhibitor Cocktail; Roche Applied Science). Proteins were separated by electrophoresis in 4–12% NuPAGE Bis–Tris gels (Invitrogen) and transferred to nitrocellulose membranes (Amersham). The antibodies used for western blotting are listed in Supplementary Table 2.

**Conditioned media.** To generate conditioned media, the cells transfected and treated as indicated were growing in serum-free growth media for 24 h. The media was filtered through a 0.45-μm filter, centrifuged at $500 \times g$ for 5 min and placed on the corresponding recipient cells for the time indicated in each experiment.

**Secretome analysis.** The CM for proteomics analysis was harvested from 6 cm plates as indicated above. CM was concentrated using Centricon (Milipore) 3 kDa filter, precipitated by TCA and washed in acetone. Protein pellets were resuspended in 6 M GndHCl in 10 mM Tris/HCl pH 8.0 with 2 mM DTT and incubated at 56 ℃ for 30 min. In-solution digestion was performed after sample dilution with 50 mM TEAB using 250 ng of LysC (Wako) (at 2 M GndHCl) and followed by 500 ng of trypsin (Promega) (at 0.6 M GndHCl). Reduced, alkylated and acidified peptides were desalted on 100 μl C18 stage tips (Thermo Fisher Scientific) and subjected to LC-MS/MS analysis.

Tryptic peptides were identified by LC-MS using an EASY-nLC 1000 (Thermo Fisher Scientific) coupled to a Q Exactive HF (Thermo Fisher Scientific) equipped with a nanoelectrospray ion source. Peptides were separated on an in-house packed column of ReproSil-Pur C18-AQ, 3 μm resin (Dr Maisch, GmbH) using a 90-min gradient of solvent A (0.5% acetic acid) and solvent B (80% acetonitrile in 0.5% acetic acid) and a flow of 250 nl/min. The mass spectrometer was operated in positive ion mode with a top 12 data-dependent acquisition, a resolution of 60,000 (at 400$m/z$), a scan range of 300–1700$m/z$ and an AGC target of 3e6 for the MS survey. MS/MS was performed at a scan range of 200–2000$m/z$ using a resolution of 30,000 (at 400$m/z$), an AGC target of 1e5, an intensity threshold of 1.0e5 and an isolation window of 1.2$m/z$. Further parameters included an exclusion time of 45 s and a maximum injection time for survey and MS/MS of 15 and 45 ms, respectively.

The raw files obtained from LC-MS were processed using the MaxQuant software 72 version 1.5.3.30. Peak lists were searched against the human UniProt database using the Andromeda search engine incorporated in MaxQuant with a tolerance level of 7 ppm for MS and 20 ppm for MS/MS[78]. Trypsin was chosen as digestion enzyme with max 2 missed cleavages allowed. Variable modifications were set to methionine oxidation, protein N-terminal acetylation, deamidation of asparagine and glutamine. Carbamidomethylation of cysteine was set as fixed modification and other parameters were kept as default.

Statistical analysis was conducted in R environment (https://www.r-project.org) using DEP (v. 1.8.0) bioconductor package for proteomics analysis, and visualizations were made using the ggplot2 R package[79]. Imputation of missing data was performed using MinProb setting based on minimal intensity values observed for each sample.

**Crystal violet staining.** Cells were seeded and reverse transfected in 6-well or 12-well plates (NUNC). 24, 48 and 72 h after transfection or treatment cells were washed twice in PBS and fixed with 10% formalin for 10 min and stained with 0.1% crystal violet solution for 30 min. Excess crystal violet stain was removed by several washes with water. The plates were allowed to dry and crystal violet was extracted by the addition of 10% acetic acid. The amount of crystal violet staining was quantified by measurement of the absorbance at 570 nm.

**Senescence-associated β-galactosidase staining.** Cells were seeded and transfected in 12-well plates (NUNC). At 72 h post-transfection they were fixed and stained using the β-galactosidase staining kit (Cell Signaling) according to the manufacturer's protocol.

**Invasion assay.** In all, 50 μl of 0.5 μg/ml Matrigel LDEV-Free(Corning) was added to the transwell (Croning). Four hundred microliters of the desired CM was placed at the bottom of the transwell with 8.0 μm pore size. In total, 50,000 MDA/MB-231 cells were resuspended in 200 μl DMEM without FBS and place on top of the transwell. After 48 h cells across the matrigel membrane were washed in PBS and fixed in cold 70% ethanol for 15 min and stained using crystal violet staining as indicated above.

**Cell fractionation.** Cells were grown in 15 cm dishes (NUNC). Nuclear/cytoplasmic fractionation was performed using Nuclei EZ Lysis Buffer (Sigma) following the manufacturer's protocol.

**Immunofluorescence.** Cells were seeded on multichamber slides (Nunc) and transfected or treated. Seventy-two hour cells were washed with PBS 1× and fixed in 4% paraformaldehyde (PFA) (Sigma) for 15 min. The cells were permeabilized using 0.1% Triton X-100 in PBS (Sigma) following antibody incubation. Images were taken using a Zeiss fluorescence microscope and ZenPro software 2011. Antibodies and dilutions are listed in Supplementary Table 2.

**Chromatin immunoprecipitation.** Cells were fixed with 1% formaldehyde for 10 min. Crosslinking was arrested by adding glycine (0.125 M) for 5 min at room temperature. The cells were subsequently harvested in SDS lysis buffer (0.5% SDS, 100 mM NaCl, 50 mM Tris-Cl pH 8.1, 5 mM EDTA pH 8.0, protease inhibitor mixture Complete [Roche], and 1 mM phenylmethylsulfonyl fluoride. Nuclei were pelleted and resuspended in IP buffer (2 volumes SDS lysis buffer:1 volume Triton-X buffer [100 mM Tris-Cl, pH 8.6, 100 mM NaCl, 5 mM EDTA pH 8.0, 5% Triton X-100]). The lysates were sonicated using BIORUPTOR sonicator for 12 cycles of 30 s and centrifuged at maximum speed. The sheared chromatin was diluted to 1 ml with IP buffer and precleared with salmon sperm DNA/recombinant protein

A agarose (Thermo Fisher Scientific) for 2 h. One per cent of the sample was used as the input control, and the remaining precleared chromatin was incubated overnight with 10 μg of antibody (see Supplementary Table 5) by incubation with salmon sperm DNA/protein A agarose (50% slurry) and centrifugation. The bead pellets were washed in low- or high-salt conditions (0.1% SDS, 1% Triton X-100, 2 mM EDTA pH 8.0, 20 mM Tris-HCl pH 8.0, and 150 mM [low]/500 mM [high] NaCl) followed by two washes with Tris-EDTA buffer. Elution buffer (0.1% SDS, 0.1 M NaHCO₃) was added to the samples and the crosslinking was reverted by incubation at 68 °C overnight. Samples were incubated 1 h at 37 °C with RNAse A (Sigma) and 45 min at 50 °C with proteinase K (Ambion). The DNA was purified using Minelute PCR Purification Kit (Qiagen) and then amplified by qPCR using primers listed in Supplementary Table 1.

**RNA sequencing and bioinformatic analysis.** RNA integrity was confirmed on an Agilent 2100 Bioanalyzer using Agilent RNA 6000 Nano kit (Agilent Technologies). RNA-seq libraries were prepared from 2 μg total RNA with CATS mRNA-seq Kit (with polyA selection) v2 x24 (Diagenode) according to the manufacturer's protocol. Concentrations of the libraries were measured using the Qubit fluorometer (Invitrogen) and fragment size was assessed on an Agilent 2100 Bioanalyzer using Agilent High Sensitivity DNA kit (Agilent Technologies). The libraries were sequenced on Illumina NextSeq500 with 75 bp single-end. Raw reads were trimmed using Cutadapt[80] to remove adapters and minimum read length after trimming was set to 18. Trimmed reads were mapped to hg38 using STAR aligner[81] (version 2.5.1a). Uniquely mapped reads were counted towards genes using featureCounts[82] (version 1.5.1). Differential expression analysis was performed with DESeq2 (ref. [83]) using FDR < 0.01. Heatmaps were generated using pheatmap package in R with default settings and relative expression as RPM calculated by Z-score scaling. GO-term analysis was carried out using PANTHER[84] (http://pantherdb.org/about.jsp, version 14.1) using Fischer's exact test with FDR multiple test correction. Only genes differentially expressed between Senescence and Senescence MIR31HG-KD and |log₂fold-change| > 0.75 were used for GO-term analysis.

**MIR31HG oligonucleotide-based pull down.** Cells were grown in 150 and 500 cm² dishes in 25 or 90 ml media, respectively, up to ~90% confluency. Prior to collection cell were irradiated with UV light at 254 nm, then scraped in ice-cold PBS, spun down at 800 × g for 5 min at 4 °C and stored at −80 °C. Amino-C12-LNA-containing oligonucleotides against MIR31HG and luciferase (five per target, Exiqon, custom design, Supplementary Table 1) were coupled to Dynabeads MyOne Carboxylic Acid (Thermo Fisher Scientific) according to the manufacturer's instructions using 1.5 nmol oligo per 100 μl beads. Prior to use, the coated beads were blocked in RNA pulldown (RP) buffer (50 mM Tris/HCl pH 7.5, 5 mM EDTA, 500 mM LiCl, 0.5% DDM, 0.2% SDS, 0.1% Na-deoxycholate, 4 M Urea, 2.5 M TCEP, protease inhibitors (Roche)) with addition of ssDNA (200 μg/ml) and BSA (1 mg/ml) and yeast RNA (200 μg/ml). In all, 0.5 g cell pellet was used per condition and resuspended in 5 ml volume of RP buffer with murine RNase inhibitor (NEB, 1:500). Lysates were sonicated in 15 ml tubes with Branson tip sonicator (2× 10sek, with 30sek break, at setting 15%) and clarified at 16,000 × g for 5 min at 4 °C. Supernatants were transferred to 2 ml tubes and pre-heated to 65 °C with shaking, then oligonucleotide-coated beads were added to lysates for 4 h incubation at 65 °C with shaking (250 μl beads/0.5 g initial cell pellet distributed in 2 ml tubes). The beads were washed four times with RP buffer at room temperature and then washed with 1 ml 50 mM TEAB buffer to remove detergents. For mass spectrometry sample preparation the beads were resuspended in 100 μl 50 mM TEAB buffer (Sigma) including 2 mM DTT. Trypsin was added (500 ng) for O/N incubation at 37 °C, then followed by treatments with DTT (10 mM) and IAA (55 mM) and acidification with TFA. One hundred microlitres StageTips (Thermo Fisher Scientific) were used for desalting and purified peptides were subjected to LC-MS analysis (as described in the above section).

Samples were analysed using instrument settings described above, and the obtained data were analysed with MaxQuant v. 1.5.2.8, using N-terminal acetylation and methionine oxidation as variable modifications. Fold change values of median peptide intensities were calculated for five replicates, and missing values were imputed using minimal intensity value detected in the analysis. Significantly enriched proteins in MIR31HG pulldown samples over control pulldown were selected based on p value < 0.05, signal intensity (fold enrichment over luciferase control) and number of unique peptides present in majority of replicates.

**Polysome profiling.** Prior to harvesting, cells were treated with 100 μg/ml cycloheximide (Sigma-Aldrich) for 3 min. Cells were washed and scrapped off in ice-cold PBS containing 100 μg/ml cycloheximide. Cell were lysed in excess of polysome lysis buffer (20 mM Tris-HCl, 150 mM KCl, 5 mM MgCl₂, 0.5% NP40 (Igepal CA-630, Sigma-Aldrich), 2 mM DTT, 100 μg/ml cycloheximide, Roche EDTA-free Protease Inhibitor (Roche) and murine RNase inhibitor (NEB)) and incubated while rotating for 10 min at 4 °C. Debris, nuclei and mitochondria were cleared by centrifugation at 12,000 × g for 15 min at 4 °C. Lysate material was normalized to equal A₂₆₀ value measured on NanoDrop. Normalized lysate, 400 μl total, was loaded on top of a 7–47% (w/v) linear sucrose gradient (Sigma) in open top polyallomer tubes (Seton Scientific) and centrifuged at 38,000 × g for 2.5 h at 4 °C

using a Beckman ultracentrifuge (Optima L-90K, Class S) with the SW40ti rotor head (Beckman). Following ultracentrifugation, gradients were fractionated by piercing the tube bottom (Brandel Piercer), pushing gradient with 60% sucrose solution at a pace of 1 ml/min while continuously measuring A₂₆₀ using the Bio-LogicP system (Bio-Rad). RNA was subsequently extracted from each fraction by QIAzol (Qiagen) and chloroform extraction. Prior to RNA extraction, 20 pg of firefly-luciferase in vitro-transcribed RNA was spiked into each fraction to control for potential loss of material in extraction protocol.

**Nascent protein labelling biotin pull-down/western blot analysis.** BJ-ER:BRAF cells 72 h were transfected using control or MIR31HG siRNAs and treated with 1 μM 4-OHT for 72 h. Cell were starved in methionine-free medium for 1 h. Subsequently, proteins were labelled with 50 μM Click-iT AHA in methionine-free medium for 4 h prior to harvesting. Click-iT Protein Reaction Buffer Kit (Thermo Fisher Scientific) and biotin alkyne (Thermo Fisher Scientific) were used to label protein according to the manufacturer. Cell pellets were resuspended in 300 μl of lysis buffer (50 mM Tris-HCl pH 8.0, 1% SDS) in the presence of protease inhibitors (Roche) and incubated 10 min on ice. Cell lysates were sonicated using Branson sonicator (3 × 10 s, 10% amplitude) and centrifuged at 15,000 × g, at 4 °C for 10 min. At least 200 μg of total protein was labelled and precipitated according to the manufacturer. Three labelling reactions were carried out for each sample to increase the final outcome. Proteins were resuspended in 50 μl lysis buffer by shaking for 10 min at 30 °C in the presence of protease inhibitors, and three labelling reactions were pooled. Sample volumes were adjusted to 1 ml with Tris-HCl pH 8.0 (final conc. 50 mM Tris-HCl, 0.1% SDS), incubated for 10 min at 30 °C and centrifuged at 15,000 × g at 25 °C for 5 min to remove undissolved material. Supernatants were incubated with 100 μl of Dynabeads™ MyOne™ Streptavidin T1 beads on a wheel rotator for 60 min at room temperature. After five washes with 1 ml of 50 mM Tris-HCl/0.1% SDS buffer proteins were eluted in 1.5× NuPAGE SDS sample buffer containing 20 mM DTT and analysed by western blot.

**MIR31HG constructs for in vitro transcription.** pGEM-MIR31HG plasmid was established by PCR amplification of MIR31HG sequence (NR_027054.2) from cDNA using primers listed in Supplementary Table 1 and cloned using TA cloning strategy with pGEM-T easy vector system kit (Promega), according to the manufacturer's protocols. MIR31HG truncations were established by PCR amplification from pGEM-MIR31HG plasmid with primers indicated in Supplementary Table 1 and cloned with In-fusion HD cloning kit (Clontech).

**In vitro transcription and labelling.** pGEM-MIR31HG WT and truncation constructs were linearized with MluI restriction enzyme. For ND4 negative control the sequence was amplified from cDNA using T7 promoter included in the forward primer sequence (Supplementary Table 1).

The in vitro transcription reaction was performed by using 1 μg of the linearized plasmid or PCR product together with 1× T7 RNA polymerase buffer, 1 mM of NTPs), 1 U/μl of RNase inhibitors murine, and 2 U/μl of T7 RNA polymerase in a total volume of 20 μl for 2 h at 37 °C.

After 30 min DNAse treatment with TURBO DNA free at 37 °C the reactions were stopped by adding 0.5 mM EDTA. Samples were mixed with formamide loading buffer and boiled 1 min at 95 °C before running on 1% agarose gel in 1× TBE. The RNA products were purified using NucleoSpin Gel and PCR Clean-up kit (Macherey-Nagel) with NTC buffer for RNA extraction. RNAs were dephosphorylated with 1 U of Calf Intestine Alkaline Phosphatase (Invitrogen) and 5′-end labelled with T4 Polynucleotide Kinase (NEB) and γ-32P (Perkin Elmer). Labelled RNAs were precipitated following phenol:chloroform extraction. Radiolabelled RNAs were checked on agarose gel in 1× TBE buffer and quantified.

**Electrophoretic mobility shift assay.** Final concentration 2 nM of radiolabelled RNA was adjusted to 5 ml with Milli-Q water, incubated 1 min at 95 °C and cooled on ice for 2 min. RNA was allowed to fold for 30 min at 37 °C in binding buffer (mM Tris-HCl pH 7.5, 50 mM KCl, 5 mM MgCl₂, 0.1 mM CaCl₂, 1 mM DTT, 0.1 mg/ml BSA, 0.4 mg/ml fragmented yeast RNA (Sigma), 5% glycerol, 0.025% bromophenol blue and 0.025% xylene cyanol). Recombinant YBX1 (Abcam) was added at the concentrations indicated and the binding reaction was carried out at 30 °C for 30 min. Samples were loaded on a non-denaturing 0.7% agarose gel in cold 1× TBE buffer. After 2 h at 120 V of gel electrophoresis, the gels were vacuum dried for 90 min at 80 °C and exposed to an Amersham Hyperfilm ECL film.

**Luciferase reporter assays.** The 3′UTR of IL1A was amplified by PCR from cDNA (see primer list) and cloned into the pGL3-promoter Firefly luciferase reporter (Promega) using XbaI restriction enzyme (primer sequences are shown in Supplementary Table 1). In total, 40,000 YBX1-GFP cells were reverse transfected with siRNAs against endogenous YBX1 in a 24-well plate. After 24 h the cells were transfected using Lipofectamine 3000 (Lifetechnologies) with 500 ng of the PGL3-promoter or PGL-3_IL1A-3′UTR together with 150 ng of pRL-Tk Renilla luciferase reporter vector for transfection control and luciferase assay normalization. Twenty-four hours after transfection fresh media containing 1 μM 4-OHT was added to the cells. Forty-eight hours after treatment Firefly and Renilla luciferase units were

measured using the Dual-Glo Luciferase Assay System (Promega) and the Glo-Max multi-detection system with Instinct software v3.1.1 (Promega).

For the translocation of NF-κB to the nucleus, 500 ng of a luciferase reporter plasmid containing four NF-κB-binding sites (Addgene Plasmid #111216) was transfected using Lipofectamine 3000 (Lifetechnologies) in BJ:ER:BRAF cells 24 h after siRNA transfection (control or siMIR31HG1-2). One hundred and fifty nanograms of pRL-Tk Renilla luciferase reporter vector was co-transfected as control and luciferase assay normalization. Twenty-four hours after transfection fresh media containing 1 μM 4-OHT was added to the cells. Forty-eight hours after treatment Firefly and Renilla luciferase units were measured using the Dual-Glo Luciferase Assay System and the Glo-Max multi-detection system with Instinct software v3.1.1(Promega).

**Proximity ligation assay**. Cells were reverse transfected and seeded in eight-well multichamber slides (Nunc Lab-Tek II Chamber Slide™ system). Twenty-four hours later media was changed and 1 μM 4-OHT was added to the corresponding wells. After 72 h cells were fixed in 4% formaldehyde for 15 min and processed for PLA using Duolink™ In Situ PLA® Probe Anti-Rabbit PLUS (Sigma) according to the manufacturer's instructions. The antibodies are listed in Supplementary Table 2. Images were obtained using a fluorescence microscope (Zeiss) and ZenPro software 2011 and quantified using Cell Profiler software v3.1.5. Five to 10 images were quantified from four independent experiments.

**In vitro kinase assay**. One microgram of recombinant YBX1 (Abcam) was incubated with 20 ng of recombinant RSK1 protein (R&D) and the corresponding in vitro-transcribed RNA in kinase buffer (25 mM TRIS/HCl, pH 7.5, 5 mM glycerolphosphate, 2 mM DTT, 10 mM MgCl₂, 50 μM ATP, 10 μCi γ-32P (Perkin Elmer), protease inhibitor mixture Complete (Roche), phosSTOP (Roche)) for 15 min at 30 °C. Laemmli buffer was added to stop the reaction and boiled at 95 °C for 5 min. Proteins were separated by electrophoresis in 4–12% NuPAGE Bis–Tris gels (Invitrogen). Subsequently, the gel was dried at 80 °C for 1 h and then exposed to an Amersham Hyperfilm ECL film. The band intensities were measured using ImageJ/Fiji software v2.0.0.

**Formaldehyde crosslinked RNA immunoprecipitation**. Cells were seeded in 15 cm plates and treated with 1 μM 4-OHT the day after for 48 h. The cells were washed twice with PBS and crosslinked with 1% formaldehyde in PBS shaking for 15 min. Crosslinking was stopped incubating with glycine 0.25 M for 5 min. Eighty to 100 mg of cell pellet was resuspended in lysis buffer (50 mM TRIS/HCl, pH 7.4, 100 mM NaCl, 0.5% Triton X-100, 5 mM EDTA, 0.25% Na-deoxycholate, Protease Inhibitor (Roche), RNase Inhibitor (NEB)) and sonicated three cycles of 10 s using a BRANSON sonicator at setting 15%. After 15 min centrifugation at 16,000 × g the supernatant was collected in a new tube. Ten per cent of the extract was saved as RNA input. The rest of the cell extract was incubated with 10 μl of GFP-Trap® Magnetic Agarose beads (Chromotek) rotating at 4 °C for 2 h. The beads were then collected using a magnetic rack and washed once with lysis buffer and five times with high-salt buffer (50 mM TRIS/HCl pH 7.4, 1 M NaCl, 0.5% Triton X-100, 1 M Urea, 5 mM EDTA, 1 mM DTT). After the last wash, the beads were resuspended in 100 μl of RIP buffer (50 mM HEPES pH 7.5, 0.1 M NaCl, 5 mM EDTA, 10 mM DTT, 0.5% Triton X-100, 1% SDS) and incubated 45 min at 70 °C to revert crosslinking. Proteins were digested incubating for 20 min at 37 °C with 10 μl proteinase K (Invitrogen, Thermo Fisher). RNA was extracted adding 600 μl of TRIzol reagent (Invitrogen, Thermo Fisher) using the manufacturer's protocol.

**Native RNA immunoprecipitation**. Cells were seeded in 15 cm plates and treated with 1 μM 4-OHT the day after for 48 h. The cells were washed twice with PBS and centrifuged at 500 × g for 5 min. The pellet was resuspended in lysis buffer (50 mM TRIS/HCl, pH 7.4, 100 mM NaCl, 0.5% Triton X-100, 5 mM EDTA, 0.25% Na-deoxycholate, Protease Inhibitor (Roche), RNase Inhibitor (NEB)). Ten per cent of the extract was saved as RNA input. The rest of the cell extract was incubated with 10 μl of GFP-Trap® Magnetic Agarose beads (Chromotek) rotating at 4 °C for 2 h. The beads were then collected using a magnetic rack and washed three times with lysis buffer. RNA was extracted adding 600 μl of TRIzol reagent (Invitrogen, Thermo Fisher) to the beads using the manufacturer's protocol.

**Reporting summary**. Further information on research design is available in the Nature Research Reporting Summary linked to this article.

## Data availability
The MS data associated with this study has been deposited to PRIDE ProteomeXchange: PXD017475. RNA-seq data have been deposited in GEO with the accession number: GSE144752. Publicly available data was downloaded from UCSC Xena platform: https://xenabrowser.net/datapages/?dataset=tcga_RSEM_gene_fpkm&host=https%3A%2F%2Ftoil.xenahubs.net&removeHub=https%3A%2F%2Fxena.treehouse.gi.ucsc.edu%3A443. https://xenabrowser.net/datapages/?dataset=mc3.v0.2.8.PUBLIC.nonsilentGene.xena&host=https%3A%2F%2Fpancanatlas.xenahubs.net&removeHub=https%3A%2F%2Fxena.treehouse.gi.ucsc.edu%3A443. The data and reagents that support the findings of

this study are available from the corresponding author upon reasonable request. Source data are available online for Figs. 1–6 and Supplementary Figs. 1–6.

## Code availability
All the codes used in this study are available upon request to the corresponding authors.

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

## Acknowledgements

We thank Morten Frodin for providing reagents and useful input for experiments. We thank Janine Erler and Kristian Helin for providing cell lines. We thank the rest of the Lund lab for contributing with helpful discussions. The Lund lab is supported by grants from the Danish Council for Independent Research (Sapere Aude program); the Novo Nordisk Foundation; and the Lundbeck Foundation and the Danish Cancer Society.

## Author contributions

M.M. and A.H.L. designed the experiments and analysed the data. M.M. wrote the manuscript and all the coauthors gave comments and input. M.M., M.L., F.S.A., B.M., and S.T. conducted experiments. F.S.A. analysed the RNA-seq data. M.L. performed the mass spectrometry experiments and analysis with help from L.M.H. and J.S.A. N.R. and A.J.S. analysed *MIR31HG* expression in THCA and CRC from TCGA public available data.

## Competing interests

The authors declare no competing interests.
