## [Peer Review File · Nature Communications]

REVIEWER COMMENTS

Reviewer #1 (Remarks to the Author):

In this manuscript, authors reported lncRNA MIR31HG which plays roles in modulating the expression and secretion of a subset of SASP components during BRAF-induced senescence. Mechanically, lncRNA MIR31HG interacts with the translation factor YBX1, facilitating its phosphorylation at serine 102 (p-YBX1S102) by the kinase RSK. p-YBX1S102 induces IL1A translation which acts as upstream regulator inducing the transcription of the other SASP mRNAs. They suggest a dual role for MIR31HG in senescence depending on its cellular localization and points to lncRNA MIR31HG as a potential therapeutic target in the treatment of senescence-related pathologies.

Overall, authors provided relatively sufficient data to support their conclusions. However, the molecular mechanism is not of sufficient depth to allow it to be fully convinced. Moreover, this is a follow up study of a previous discovery, therefore the novelty aspects are much diminished. Based on all these, this manuscript, at least this version, does not meet the journal's required standard.

Major concerns:

1. Northern blot or RACE assay should be performed to determine the full length of MIR31HG and its possible isoforms. Authors should also design experiments to exclude the possible coding capacity of MIR31HG. In another word, they should prove that MIR31HG is a bona fide noncoding RNA.
2. Figure 2B, β -galactosidase graphs need to provide quantitative information.
3. Figure 3B, 4G, ChIP data should be presented as % of input, which gives readers a better idea of the level of enrichment.
4. The intensity of protein bands should be quantified by densitometry analysis (Figure 1H, 2A, 3A, 3E, 4F, 5A, 5B, 5C, 5D and 6A).
5. The microscope picture needs to be scaled (Figure 2B, 2D, 3C, 3D, 6B).
6. Why only IL6 protein expression was analyzed by western blot in figure 1H? what about CXCL1 (Figure 1H)?
7. The interaction of MIR31HG and YBX1 is one of the key results of this paper. But there is too little evidences. Western blot should be provided to verify the results from mass spectrum. An in vitro assay should be provided to demonstrate that MIR31HG indeed directly binds with YBX1. And using RNA-FISH and IF to confirm the spatial basis of co-localization for these two factors.
8. It was previously reported that silencing of MIR31HG could induce cell senescence by the upregulation of p16. And p53, p27 and p16 are all important mediators of senescence. In Figure 2A, authors only showed the change of the senescence marker p53. Other factors should also be examined.
9. In Supplementary Figure 2 A, C, the statistical significance should be included.
10. MIR31HG knock-down in senescent cells reduced CEBPB protein expression and activated p-RELA. But the level of CEBPB is inconsistent in control cells (Figure 3A). Authors should explain this.
11. Why the changes of p53 and p16 caused by two independent shRNAs exhibit opposite effect in Supplementary Figure 4 B?
12. The protein level of RSK should be showed in Figure 5A, B. And authors should detect the protein level of RKS and p-RSK in Figure 5C.
13. A recent article (J Cancer. 2020; 11(6): 1457–1467) reported that p16 can affect the stability of IL1A mRNA. Does this jeopardize your conclusion?
14. Data presented in Supplement Figure 5B is weird, could it be that the second column and the third column are messed up?
15. Without 4-OHT, is YBX phosphorylation regulated by RSK?
16. Authors should design deletion-mapping experiments to investigate the structural basis of the interactions between MIR31HG and RSK or YBX.

17. To further analyze the role of MIR31HG in the interaction of YBX1 with its kinase RSK, authors performed proximity ligation assays (PLA) using antibodies to explore the role of MIR31HG in the interaction of YBX1 and RSK (Figure 6B). This is another key result for this study that requires multiple experimental approaches to strengthen this conclusion. Authors are suggested to utilize additional techniques such as immunoprecipitation or mammalian Two-Hybrid assays.

Reviewer #2 (Remarks to the Author):

In this report, the Lund laboratory continues to investigate the role of the long noncoding RNA MIR31HG in the regulation of senescence. While in the nucleus MIR31HG represses p16 expression to enable proliferation (as the group reported in 2015), the export of MIR31HG to the cytoplasm has other consequences on senescence that are explored in this study – specifically the senescence-associated secretory phenotype (SASP). Using BRAF as a model trigger of senescence, the authors find that cytoplasmic MIR31HG promotes the expression and secretion of a subset of SASP components by interacting with the translation factor YBX1, and thereby facilitating its phosphorylation (p-YBX1S102) by the kinase RSK. p-YBX1S102 induces IL1A translation, and IL1A in turn induces the transcription of other SASP mRNAs.

This is an interesting report that expands upon the role of lncRNA MIR31HG on cellular senescence. With growing interest in identifying ways to modify specific segments of the senescence program, this report is exciting in that it highlights an upstream mediator of the SASP, believed to be a harmful component of the senescent phenotype. There are a few aspects of the study that must be addressed before the authors' model is fully supported.

Main Comments

1. Additional support is needed for the sequence of events proposed by the authors. If IL1A is the direct initial regulator of the SASP paradigm (MIR31HG → p-YBX → IL1A translation → other SASP factors), the authors need to gain evidence that IL1A functions in an autocrine/paracrine manner. Use of neutralizing antibodies to block IL1A actions from outside of the cell would allow the authors to test the hypothesis that the newly translated IL1A initiates (from the extracellular space) a downstream wave of SASP factor production.

2. Besides the biochemical measurement of MIR31HG, it is critical that the authors (1) employ RNA-FISH or another in situ visualization method that MIR31HG increases in the cytoplasm of senescent cells, and (2) measure copy number in the nucleus and cytoplasm using digital PCR or another quantification method. Points (1) and (2) are complementary and will give the authors insight into the number of MIR31HG molecules in the cytoplasm of senescent cells in this paradigm.

3. Related to point 2, what is the stoichiometry of YBX to MIR31HG? Are their molecular ratios consistent with the effects seen for YBX phosphorylation?

4. In order to establish if phosphorylation of YBX at Ser-102 requires the interaction of YBX with MIR31HG (another point in the model), the authors must map where on MIR31HG YBX binds and see if a mutant MIR31HG lacking this site (i) does not bind YBX, (ii) does not phosphorylate YBX, (iii) does not increase IL1A translation, and (iv) does not elevate SASP factors.

5. How universal is this effect? Besides OIS, does MIR31HG raise IL1A levels and hence the SASP in

other models of senescence?

Minor points:

- Formaldehyde-induced crosslinking followed by ribonucleoprotein immunoprecipitation is best abbreviated CLIP (not RIP)
- Further discussion on how MIR31HG might promote the interaction between RSK and RBX1 would be helpful.

Reviewer #3 (Remarks to the Author):

The manuscript by Marta Montes and Professor Anders H Lund's team focused on the study of lncRNA MIR31HG to understand how it regulates the senescence associated secretory phenotype. This work is a continue study after they published their work about lncRNA MIR31HG in 2015. Here they included secretome and RNA-binding proteomics to identify the key players with lncRNA MIR31HG. This work is interesting and provides the useful information, however, some issues need to be confirmed and clarified as suggested below before publication.

1. Page 5, please change "TGCA data" to "TCGA data".
2. The top 1 and 2 of GO categories in Supplementary Fig. 1A are Receptor activity and microtubule binding, did the authors further validate the functions? Why did the authors choose not the top 5 functions as the further study?
3. What is the expression level of ICAM1 after MIR31HG knock-down in TIG3 ER:BRAF cell line?
4. I suggest the authors deposit the secretome and RNA-binding proteome data as well as transcriptome into public proteome and transcriptome databases such as ProteomeXchange and Gene Expression Omnibus (GEO).
5. In page 6, The sentence described the results "Among the less secreted proteins upon MIR31HG depletion were inflammatory SASP factors IL6 and CXCL1 (Figure 1F, and Supplementary Fig.1D and 1E).", however, no CXCL1 data was shown in Figure 1H, Supplementary Fig.1D and 1E. How were the validation results using qRT-PCR and western blot analyses? I think the conclusion "MIR31HG knock-down reduces the production of several SASP components during OIS already at the transcriptional level and these changes are reflected in the secretome." is too strong since only IL6 has the consistent results in different assays. I suggest the author need to provide some evidence to strengthen the conclusions.
6. I suggest the authors showed the bar chart for Figure 2A and 2B.
7. The authors described "NF-kB translocation to the nucleus was inhibited upon MIR31HG depletion (Figure 3C).", however, I could not see the conclusion in the figure. In Figure 3C, I only see the NF-kB subunit RELA in MIR31HG depletion cells was distributed strongly in the cytosol compared to Control. I suggest the authors use cellular fractionation method to isolate the nuclear and cytosol proteins and further performed Western blot to see whether NF-kB level in nucleus is higher than in cytosol.
8. In page 9, the authors described "MIR31HG knock-down does not affect mTOR signalling activation (Supplementary Fig. 3H)", in the data, they didn't detect the protein level of mTOR, why did they conclude that?

9. In page 10, the authors described "IL1A protein levels were strongly reduced (Figure 4F) whereas the mRNA level remained unaltered (Figure 4D).", I am curious how to explain the inconsistent results?

10. I suggest the authors use some prediction tools to predict the predicted binding motif of interacting proteins on MIR31HG sequence to see whether the proteins identified by mass spectrometry have direct binding sites with MIR31HG. Additionally, I am curious where the binding sites are located in the MIR31HG-YBX1 interaction.

11. When I compared the data in the manuscript with their previous published paper "Montes, M. et al. The lncRNA MIR31HG regulates p16(INK4A) expression to modulate senescence. Nature communications 6, 6967 (2015).", the IL1A expression level has no significant change in the manuscript (Supp. Figure 1B), however, has significant change in the previous paper (Figure 3C). How to explain the inconsistent results?

12. In the MATERIALS AND METHODS section "MIR31HG expression analysis in tumor samples", I suggest the authors provide the data numbers they used.

13. The authors must correct the words and insert space between numbers and units such as "200ug/mL", "1mg/mL", "15mL" into "200 ug/mL", "1 mg/mL", "15 mL"...

Reviewer #4 (Remarks to the Author):

Montes and colleagues show that MIR31HG regulates the SASP during BRAF-induced senescence by facilitating YBX1 phosphorylation at serine 102 by the kinase RSK. They connect this event to IL1A translation which is a well-known regulator of the SASP. This is an important and novel study that uncovers a new regulatory axis for the SASP. However, the results supporting a translational control are not strong at this point. The following points require authors clarification.

1- In Figure 2a, the induction of p53 is not well characterized. P53 target genes should be part of this analysis.

2- I do not think the results presented in Fig 3H and quantified in I-J support the authors claim that IL1A is regulated at the translational level. A 25% reduction at the most cannot explain the dramatic results reported in figure 3E and F. Since mRNA levels do not change other mechanisms should be considered including, protein stability, mRNA localisation and miRNAs.

3- The biological meaning found in this paper is an inhibition of the SASP and its ability to induce invasion when MIR31HG is depleted. To validate the reported mechanisms authors should present similar biological data when analyzing the role of YBX1 phosphorylation and Rsk in BRAF-induced senescence.

4- Mechanistically figure 5E supports the translational control model. Can the authors repeat their polysome profile analysis in this setting? Also the identification of RNA motifs mediating the regulation will further clarify the translational control model.

Minor points:

1- Line 50 : ... and the presence OF senescence-associated heterochromatin...

2- Figure 2B needs quantification and a magnification bar.

3- Wajapeyee and al. (Cell, 2008) have shown that IGFBP7 was induced and secreted following BRAFV600E expression and that it could induce senescence and apoptosis in an autocrine/paracrine fashion. Their results show that IGFBP7 is not induced upon BRAFV600E induction (Fig. 1D and fig. 4C), can you comment on that? Also, since IGFBP7 is known to be involved in paracrine senescence,

and that when you knock down MIR31HG you induce IGFBP7 at the mRNA level (Fig. 1D and fig. 4C)
is it more secreted? If so, could it have an impact on the invasion phenotype in figure 2

4- No scale bars: Fig. 3C, 3D and Fig. 6B

5- Fig. 6B: can you provide a control for the PLA using only one of the primary antibodies and both probe-conjugated secondary antibodies. These controls ensure that the signal is really due to the binding of the two antibodies in proximity.

REVIEWER COMMENTS

Reviewer #1

In this manuscript, authors reported lncRNA MIR31HG which plays roles in modulating the expression and secretion of a subset of SASP components during BRAF-induced senescence. Mechanically, lncRNA MIR31HG interacts with the translation factor YBX1, facilitating its phosphorylation at serine 102 (p-YBX1S102) by the kinase RSK. p-YBX1S102 induces IL1A translation which acts as upstream regulator inducing the transcription of the other SASP mRNAs. They suggest a dual role for MIR31HG in senescence depending on its cellular localization and points to lncRNA MIR31HG as a potential therapeutic target in the treatment of senescence-related pathologies.

Overall, authors provided relatively sufficient data to support their conclusions. However, the molecular mechanism is not of sufficient depth to allow it to be fully convinced. Moreover, this is a follow up study of a previous discovery, therefore the novelty aspects are much diminished. Based on all these, this manuscript, at least this version, does not meet the journal's required standard.

Major concerns:

1. Northern blot or RACE assay should be performed to determine the full length of MIR31HG and its possible isoforms. Authors should also design experiments to exclude the possible coding capacity of MIR31HG. In another word, they should prove that MIR31HG is a bona fide noncoding RNA.

Using the data obtained from the polysome profiles in BJ ER:BRAF treated with 1 μ M 4-OHT in main Figure 3 we have quantified the amount of *MIR31HG* RNA present in each of the polysome fractions by RT-qPCR. We have seen that distribution of *MIR31HG* RNA resembled more the profile of the long non-coding *MALAT1* than the protein-coding *ACTB* mRNA with less accumulation in the polysome fractions (11-15) (Fig A). Specially the amount of *MIR31HG* in the heavy fractions (13-15) was significantly lower than *ACTB* (Fig B). Many lncRNAs are found in ribosomes without being translated, likely due to interaction with mRNA or with proteins present in the polysome fractions (Carlevaro-Fita, Rahim et al. 2016). For instance, YBX1, *MIR31HG* interaction partner, is found in the polysomes (Skabkin, Kiselyova et al. 2004, Tanaka, Ohashi et al. 2010). This could explain the residual levels of *MIR31HG* detected in the polysomes.

(A) Percentage of *ACTB*, *MIR31HG* and *MALAT1* RNA in the difference fractions. Fractions 11-15 correspond to the polysome fractions. Red area shows the heavy polysome fractions and grey area shows the light polysome fractions. (B) Amount of *MIR31HG* and *MALAT1* RNA relative to *ACTB* (as set to 1) present in the heavy polysome fractions. All statistical significances were calculated using two-tailed Student t-tests, **p < 0.01; ****p < 0.0001. All error bars represent means ± s.d. n=3.

2. Figure 2B, β -galactosidase graphs need to provide quantitative information. Quantification has been provided.

3. Figure 3B, 4G, ChIP data should be presented as % of input, which gives readers a better idea of the level of enrichment. Changes have been made.

4. The intensity of protein bands should be quantified by densitometry analysis (Figure 1H, 2A, 3A, 3E, 4F, 5A, 5B, 5C, 5D and 6A). All western blots have been quantified.

5. The microscope picture needs to be scaled (Figure 2B, 2D, 3C, 3D, 6B). Scale bars have been included in the figures.

6. Why only IL6 protein expression was analyzed by western blot in figure 1H? what about CXCL1 (Figure 1H)?

We tried to detect CXCL1 by western blot of precipitated proteins from the CM, as we did for IL6, without success. However, due to the high sensitivity of the ELISA assay we were able to detect CXCL1 and now, other SASP components (IL8 and MMP3) that were undetected by western blot. These new results have been incorporated to the main text in page 6 and are presented in the main Figure 1g. The validation of IL6 by western blot has been moved from Figure 1h to Supplementary Fig. 1I to avoid confusion.

BJ ER:BRAF (Control or si*MIR31HG*, 1-2) were treated for 72 h with ethanol (Control) or 1 μ M 4-OHT. The CM was then harvested and secreted IL8 (A), MMP3 (B) and CXCL1(C) were measured

by ELISA. The graphs show the pg/ml of secreted protein (n>3). All statistical significances were calculated using two-tailed Student t-tests, *p< 0.05; **p < 0.01; error bars represent mean ± s.d.

7. The interaction of MIR31HG and YBX1 is one of the key results of this paper. But there is too little evidences. Western blot should be provided to verify the results from mass spectrum. An *in vitro* assay should be provided to demonstrate that MIR31HG indeed directly binds with YBX1. And using RNA-FISH and IF to confirm the spatial basis of co-localization for these two factors.

We have validated the mass spectrometry results by performing RIP assays using GFP-YBX1 overexpressing cell lines and detecting by RT-qPCR the interaction with *MIR31HG* (manuscript Figure 4h). We have not succeeded in the detection MIR31HG by RNA-FISH using different techniques to be able to perform co-localization studies. To further validate the interaction, we have performed *in vitro* binding assays and observed that YBX1 is able to bind MIR31HG *in vitro* (see point 16). In electrophoretic mobility-shift assay (EMSA) experiments incubating the corresponding *in vitro* transcribed RNA with increasing amounts of recombinant YBX1 (0, 100, 200, 400 ng). *MIR31HG* -WT resulted in a band shift when incubated with increasing amounts of YBX1 indicating the binding, whereas *ND4*, a mitochondrial transcript previously used as a negative control (manuscript, Figure 4h) did not bind (see point 16 for the complete experiment including MIR31HG deletion mutants and competition assays). These results have been included in figure 4 in the manuscript.

8. It was previously reported that silencing of MIR31HG could induce cell senescence by the upregulation of p16. And p53, p27 and p16 are all important mediators of senescence. In Figure 2A, authors only showed the change of the senescence marker p53. Other factors should also be examined.

In our previous work, we demonstrate that MIR31HG regulates senescence through p16. The cellular model we used is TIG3-BRAF cells, where p16 is a crucial regulator of cellular senescence. We have also seen upregulation of p16 in BJ-BRAF cells, however the expression of p16 is not as clear as in TIG3. Here, we have measured the levels of p16 but the levels were not detectable in most of the replicates. As markers for cellular senescence we observed an increased β-galactosidase activity and now, we provide further analysis of cellular growth and p53 response.

We have measured the proliferation of BJ and IMR90 fibroblasts that had been in contact with the conditioned media (CM) from BJ ER:BRAF cells (Control or si*MIR31HG*1-2) treated with ethanol

(Control) or 1 μ M 4-OHT for 72 h. BJ or IMR90 cells growing in the CM from senescence-treated cells proliferated less than cells in contact with CM from control cells, independently of the presence or absence of *MIR31HG* (Fig. A and B). Furthermore, we observed a lower gene expression of a subset of proliferation-related genes in BJ and IMR90 cultured in CM from senescent cells (Fig. C and D). BJ cells in contact with CM from 4-OHT treated cells showed an increase p53 protein levels. This increase was observed in *MIR31HG*-depleted cell as well (manuscript, Figure 2a). To validate this result in another cell line, we analysed p53 protein expression in IMR90 cells in contact with the CM described above. As we have previously seen for BJ cells, p53 was increased in all the senescence conditions but not in control-treated cells, independently of *MIR31HG* expression (Fig. E). The expression level of a subset of p53- target genes (Fischer 2017) was also upregulated in BJ and IMR90 cultured in CM from senescent cells in the presence and absence of *MIR31HG* (Fig. F and Fig. 2G), corroborating the upregulation of the p53 response. These results have been incorporated to the manuscript in page 7. The relative growth in BJ cells is now shown in main Figure 2d. The rest of the data is presented in Supplementary Fig. 2 (a-f).

(A-B) Wild type BJ (A) and IMR90 (B) cells were incubated for 72 h with the CM collected from BJ ER:BRAF cells (Control or si*MIR31HG*1-2) treated with ethanol (Control) or 1 μ M 4-OHT for 72 h. The number of cells was measured by crystal violet staining dissolved in acetic acid and measure at 590 nm. The graph shows the absorbance relative to the control cells set as 1 (n=3). **(C-D)** qRT-PCR analysis of a subset of cell-cycle related genes normalized to housekeeping genes (*HPRT1* and *RPLP0*) in total RNA extracted from cells described in (A-B). The graph shows the RNA expression relative to control cells set to 1 (n=3). **(E)** IMR90 cells were incubated for 72 h with the CM collected

from BJ ER:BRAF cells (Control or siMIR31HG1-2) treated with ethanol (Control) or 1 μ M 4-OHT for 72 h. Left, whole cell extracts were analysed by western blot for p53 and Vinculin. Molecular weight marker is shown in kDa. Right, quantification of the intensity of the bands using Fiji software (n=3). (F-G) qRT-PCR analysis of a subset of p53-target genes normalized to housekeeping genes (*HPRT1* and *RPLP0*) in total RNA extracted from cells described in (A-B). All statistical significances were calculated using two-tailed Student t-tests, *p< 0.05; **p < 0.01; ***p<0.001. All error bars represent means \pm s.d. n=3.

9. In Supplementary Figure 2 A, C, the statistical significance should be included.

We have included the statistical significance in the figure.

10. MIR31HG knock-down in senescent cells reduced CEBPB protein expression and activated p-RELA. But the level of CEBPB is inconsistent in control cells (Figure 3A). Authors should explain this.

The western blot shown in Figure 3a shows a representative experiment. We have now included the quantification of the intensity of the bands from three independent experiments in Figure 3b. No significant difference was observed in the levels of CEBPB in control cells.

11. Why the changes of p53 and p16 caused by two independent shRNAs exhibit opposite effect in Supplementary Figure 4 B?

We have analysed this issue and repeated the experiments also including a new siRNA against YBX1. We now show that YBX1 KD decreases proliferation (Fig. A) although it also shows reduced levels of p53 (Fig. B). p21 was slightly upregulated in YBX1 KD cells and p16 was inconsistent and not detected in all the experiments (Fig. B). Therefore, it has been excluded from the figure. The mechanism behind the role of YBX1 in proliferation remains to be elucidated, which we believe is out of the scope for this paper. We have replaced Supplementary Figure 4a and 4b with the new data including the extra siRNA.

(A) BJ ER:BRAF (control or siYBX1-1-2-3) were stained with crystal violet staining to address the cell growth at the indicated days post-transfection. The graph shows the absorbance (590 nM) measured after dissolving the crystal violet in 10% acetic acid (n=3). (b) BJ ER:BRAF (control or siYBX1-1-2-3) were analysed for western blot 72 h post-transfection for YBX1, p53, p21 and Vinculin. Molecular weight marker is shown in kDa (n=2).

12. The protein level of RSK should be showed in Figure 5A, B. And authors should detect the protein level of RKS and p-RSK in Figure 5C.

We have included p-Rsk and RSK levels in Figure 5.

13. A recent article (J Cancer. 2020; 11(6): 1457–1467) reported that p16 can affect the stability of IL1A mRNA. Does this jeopardize your conclusion?

When we knock-down *MIR31HG* we do not observe any change in *IL1A* mRNA stability - just in protein expression. To corroborate this result, we have knocked-down p16 and analysed by qRT-PCR the mRNA levels of several SASP components, including *IL1A*. The mRNA levels of the genes measured showed no difference in expression upon depletion of p16, indicating that in our model, p16 does not have a role in *IL1A* mRNA stability.

14. Data presented in Supplement Figure 5B is weird, could it be that the second column and the third column are messed up? We have corrected this figure.

15. Without 4-OHT, is YBX phosphorylation regulated by RSK?

The levels of p-YBX1 and p-RSK are very low in control proliferating cells, however, when we deplete RSK with siRNAs we can observe that also p-YBX1 levels are decreased indicating that some degree of regulation occurs in control cells.

16. Authors should design deletion-mapping experiments to investigate the structural basis of the interactions between MIR31HG and RSK or YBX.

We did not detect interaction between *MIR31HG* and RSK via RIP (Main manuscript, Supplementary Fig. 6b) or *in vitro* binding assays (data not shown), suggesting that the transcript is not acting as a scaffold bridging together the kinase and its substrate. In order to map the interactions requested, we have focused in identifying the region in *MIR31HG* mediating YBX1 binding.

Based on iCLIP studies (Goodarzi, Liu et al. 2015, Wu, Fu et al. 2015), we performed an *in silico* prediction of YBX1 binding sites on *MIR31HG* sequence and identified 5 putative binding sites (BS) (Fig. A and B). We generated *MIR31HG* mutants harbouring deletions including the predicted binding sites (Fig. C). To identify functional YBX1 binding sites on *MIR31HG*, we performed electrophoretic mobility-shift assay (EMSA) experiments incubating the corresponding *in vitro* transcribed RNA with recombinant YBX1. The full length *MIR31HG* (WT) resulted in a band shift when incubated with increasing amounts of YBX1 indicating the binding, whereas *ND4*, a mitochondrial transcript previously used as a negative control (manuscript, Figure 4h) did not bind (Fig. D). Mutant 1 and 5 (Mut1 and Mut5) bind YBX1 to the same extent as the WT whereas mutants 2, 3 and 4 (Mut2, Mut3 and Mut4) did not bind (Fig. D). These results indicate that the predicted BS3 is the responsible for the binding of YBX1 to *MIR31HG*. To validate these results, we performed competition assays incubating ³²P-labelled *MIR31HG* WT transcript with increasing amounts of unlabelled Mut5 or Mut3. We observed that only Mut5 competed for the binding of YBX1 (Fig. E). These results have been incorporated to the manuscript in page 12 and to main Figure 4 (i-k).

(A) Binding motifs found in published iCLIP data. **(B)** Representation of the *MIR31HG* sequence with putative YBX1 binding sites (BS). Putative BS of YBX1 from iCLIP data were mapped on *MIR31HG* RNA sequence. BS1-3 were found in Goodarzi *et al.* iCLIP data and BS5-4 were retrieved from Wu *et al.* dataset. **(C)** Representation of *MIR31HG* truncations (mut1 to mut5) where different BS were removed from the RNA sequence. **(D)** Electrophoretic mobility assay showing the in vitro binding of 2 nM of the corresponding radiolabelled transcript and the indicated amounts of recombinant YBX1. BSA at the highest concentration (400ng) was used as a negative control. **(E)** Electrophoretic mobility assay showing the in vitro binding of 400 ng of YBX1 to 2 nM of the radiolabelled *MIR31HG* WT in competition with the indicating amounts of unlabelled Mut1 or Mut3.

17. To further analyze the role of *MIR31HG* in the interaction of YBX1 with its kinase RSK, authors performed proximity ligation assays (PLA) using antibodies to explore the role of *MIR31HG* in the interaction of YBX1 and RSK (Figure 6B). This is another key result for this study that requires multiple experimental approaches to strengthen this conclusion. Authors are suggested to utilize additional techniques such as immunoprecipitation or mammalian Two-Hybrid assays.

We have tried to validate the PLA results by co-immunoprecipitation techniques using different crosslinking conditions to be able to detect RSK and YBX1 interaction in the presence or absence of *MIR31HG* without success. As mentioned by reviewer 1, we could in addition have tried yeast-two hybrid assays, but the use of a bacterial system does not allow us to modulate the expression of *MIR31HG*, which is the relevant part of this study presented here. As previously reported, YBX1 is an already known substrate for RSK (Stratford, Fry et al. 2008). To further corroborate this interaction, we have performed in vitro kinase assay using recombinant RSK and recombinant YBX1 and observed that YBX1 is phosphorylated in a RSK-dependent manner. Interestingly, RSK auto-phosphorylates itself, consistent with previous publications (Roux, Richards et al. 2003) (Fig. A). In this setup, we have added *in vitro* transcribed *ND4*, previously used as negative control for YBX1 binding, a WT *MIR31HG* and a mutant lacking YBX1 BS3 (mut3). Interestingly we observed the highest phosphorylation levels of YBX1 in the presence of *MIR31HG* (Fig. B). These results supporting the PLA assay have been included in the manuscript (pages 15-16) and are shown in main Figure 6e.

(A) In vitro kinase assay using recombinant YBX1 (1 μ g) and increasing amounts of recombinant RSK1. **(B)** In vitro kinase assay using recombinant YBX1 (1 μ g) and recombinant RSK1 (20 ng) in the presence of the indicated in vitro transcribed RNAs. The graph shows the p-YBX1 band quantification related to p-RSK1. * $p < 0.05$; ** $p < 0.01$; error bars represent mean \pm s.d.

Reviewer #2 (Remarks to the Author):

In this report, the Lund laboratory continues to investigate the role of the long noncoding RNA MIR31HG in the regulation of senescence. While in the nucleus MIR31HG represses p16 expression to enable proliferation (as the group reported in 2015), the export of MIR31HG to the cytoplasm has other consequences on senescence that are explored in this study – specifically the senescence-associated secretory phenotype (SASP). Using BRAF as a model trigger of senescence, the authors find that cytoplasmic MIR31HG promotes the expression and secretion of a subset of SASP components by interacting with the translation factor YBX1, and thereby facilitating its phosphorylation (p-YBX1S102) by the kinase RSK. p-YBX1S102 induces IL1A translation, and IL1A in turn induces the transcription of other SASP mRNAs.

This is an interesting report that expands upon the role of lncRNA MIR31HG on cellular senescence. With growing interest in identifying ways to modify specific segments of the senescence program, this report is exciting in that it highlights an upstream mediator of the SASP, believed to be a harmful component of the senescent phenotype. There are a few aspects of the study that must be addressed before the authors' model is fully supported.

Main Comments

1. Additional support is needed for the sequence of events proposed by the authors. If IL1A is the direct initial regulator of the SASP paradigm (MIR31HG \rightarrow p-YBX \rightarrow IL1A translation \rightarrow other SASP factors), the authors need to gain evidence that IL1A functions in an autocrine/paracrine manner. Use of neutralizing antibodies to block IL1A actions from outside of the cell would allow

the authors to test the hypothesis that the newly translated IL1A initiates (from the extracellular space) a downstream wave of SASP factor production.

In the previous version of Supplementary Fig. 3C, we show the ability of adding exogenous recombinant IL1A to induce the SASP. To strengthen this issue, we knocked down IL1A in senescent cells and use this CM to assess the expression of SASP components by qRT-PCR in recipient BJ and IMR90 cells. This experiment provides equivalent results to utilizing neutralizing antibodies that requires more optimization. We have observed that CM from IL1A knock-down senescent cells was not able to induce paracrine expression of SASP components in either BJ or IMR90 cells (Fig. A and B). We have included these results in Supplementary Fig. 3g.

Wild type BJ (A) and IMR90 (B) cells were incubated for 72 h with the CM collected from BJ ER:BRAF cells (Control, siIL1A or siMIR31HG) treated with ethanol (Control) or 1 μ M 4-OHT for 72h. The graphs show the RNA expression of SASP components relative to housekeeping genes (*HPRT1* and *RPLP0* relative to control ethanol-treated cells set to 1 (n=2).

2. Besides the biochemical measurement of MIR31HG, it is critical that the authors (1) employ RNA-FISH or another in situ visualization method that MIR31HG increases in the cytoplasm of senescent cells, and (2) measure copy number in the nucleus and cytoplasm using digital PCR or another

quantification method. Points (1) and (2) are complementary and will give the authors insight into the number of *MIR31HG* molecules in the cytoplasm of senescent cells in this paradigm.

We have measured the number of *MIR31HG* molecules in the cytoplasm and nucleus from control and 4-OHT treated (senescence) cells by qPCR and comparison to a dilution series of an *in vitro* transcribed *MIR31HG*. These results showed that the absolute number of *MIR31HG* molecules decreases in the nucleus, whereas it increases in the cytoplasm in senescence.

3. Related to point 2, what is the stoichiometry of YBX to *MIR31HG*? Are their molecular ratios consistent with the effects seen for YBX phosphorylation?

We are not able to say anything about the stoichiometry. *MIR31HG* is very lowly expressed compared to the high levels of YBX1. We believe the binding of *MIR31HG* to YBX1 acts like a “platform” that allows RSK to phosphorylate YBX1. We speculate this to occur in a processive manner that would not require equimolar amounts of the interactors.

4. In order to establish if phosphorylation of YBX at Ser-102 requires the interaction of YBX with *MIR31HG* (another point in the model), the authors must map where on *MIR31HG* YBX binds and see if a mutant *MIR31HG* lacking this site (i) does not bind YBX, (ii) does not phosphorylate YBX, (iii) does not increase *IL1A* translation, and (iv) does not elevate SASP factors.

Based on iCLIP studies (Goodarzi, Liu et al. 2015, Wu, Fu et al. 2015), we performed an *in silico* prediction of YBX1 binding sites on *MIR31HG* sequence and identified 5 putative binding sites (BS) (Fig. A and B). We generated *MIR31HG* mutants harbouring deletions including the predicted binding sites (Fig. C). To identify functional YBX1 binding sites on *MIR31HG*, we performed electrophoretic mobility-shift assay (EMSA) experiments incubating the corresponding *in vitro* transcribed RNA with recombinant YBX1. The full length *MIR31HG* (WT) resulted in a band shift when incubated with increasing amounts of YBX1 indicating the binding, whereas *ND4*, a mitochondrial transcript previously used as a negative control (manuscript, Figure 4h) did not bind (Fig. 4D). Mutant 1 and 5 (Mut1 and Mut5) bind YBX1 to the same extent as the WT whereas mutants 2, 3 and 4 (Mut2, Mut3 and Mut4) did not bind (Fig. D). These results indicate that the predicted BS3

is the responsible for the binding of YBX1 to *MIR31HG*. To validate these results, we performed competition assays incubating 32 P-labelled *MIR31HG* WT transcript with increasing amounts of unlabelled Mut5 or Mut3. We observed that only Mut5 competed for the binding of YBX1 (Fig. E). These results have been incorporated to the manuscript in page 12 and to main Figure 4 (i-k).

In order to assess the impact of lost interactions in the SASP expression, we created BJ ER:BRAF expressing doxycycline-inducible *MIR31HG* constructs (WT, Mut5 and Mut3) (Fig. F). Interestingly, overexpression of WT and Mut5 transcripts showed increased levels of p-YBX1 compared to an empty cell line and Mut3 overexpression (unable to bind *MIR31HG*) (Fig. G and H), confirming that the interaction between *MIR31HG* and YBX1 facilitates its phosphorylation.

We were not able to see consistent changes in IL1A protein expression levels in this setting. The residual levels of pYBX1 might be responsible of facilitating low levels of IL1A translation and therefore, moderate SASP response. This could generate a feedback loop resulting in high IL1A expression levels and SASP induction.

These results have been incorporated to the manuscript (page 16) and to Figure 6f,g and to Supplementary Fig. 6d.

(A) Binding motifs found in published iCLIP data. (B) Representation of the *MIR31HG* sequence with putative YBX1 binding sites (BS). Putative BS of YBX1 from iCLIP data were mapped on *MIR31HG* RNA sequence. BS1-3 were found in Goodarzi *et al.* iCLIP data and BS5-4 were retrieved from Wu *et al.* dataset. (C) Representation of *MIR31HG* truncations (mut1 to mut5) where different

BS were removed from the RNA sequence. **(D)** Electrophoretic mobility assay showing the in vitro binding of 2 nM of the corresponding radiolabelled transcript and the indicated amounts of recombinant YBX1. BSA at the highest concentration (400ng) was used as a negative control. **(E)** Electrophoretic mobility assay showing the in vitro binding of 400 ng of YBX1 to 2 nM of the radiolabelled *MIR31HG* WT in competition with the indicating amounts of unlabelled Mut1 or Mut3. **(F)** qRT-PCR analysis for *MIR31HG* expression levels of the doxycycline-inducible cell lines overexpressing *MIR31HG* WT, Mut3 and Mut5 compared to an empty cell lines set to 1. **(G)** BJ ER:BRAF cells expressing a doxycycline-induced empty, *MIR31HG* WT, Mut3 or Mut5 treated for 36 h with ethanol (Control) or 1 μ M 4-OHT. Total extracts were analysed by western blot for p-YBX1, total YBX1 and Vinculin. **(H)** Quantification of the p-YBX1 band intensities relative to total YBX1 and Vinculin.

5. How universal is this effect? Besides OIS, does *MIR31HG* raise IL1A levels and hence the SASP in other models of senescence?

To investigate a broader effect of *MIR31HG* in another senescence model, we assessed doxorubicin-induced senescence in BJ cells. As expected, cells showed senescence markers after 48 h treatment with 500 nM doxorubicin as evidenced by a reduced cell growth, quantified by crystal violet staining (Fig. A) or the presence of beta-galactosidase positive cells (Fig. B). Doxorubicin induced the expression of *MIR31HG* and also SASP components such as IL6, IL8 or CXCL1 (Fig C). *p21* mRNA was also induced consistently with a decreased cell proliferation (Fig. C). Interestingly, depletion of *MIR31HG* with two different siRNAs resulted in decreased expression of SASP components but not *p21* (Fig. C). These results support the role for *MIR31HG* in the regulation of part of the SASP in different types of senescence and they have been included in the manuscript (page 6) and are shown in supplementary Fig. 1c-e.

(A) Representative image of a crystal violet staining experiment in BJ cells treated with DMSO (control) or with 500 nM doxorubicin for 48 h. (B) Representative images of the β -galactosidase staining in BJ cells treated with DMSO (control) or with 500nM doxorubicin for 48 h. Scale bar: 50 μ m. (C) qRT-PCR analysis of selected components of the SASP normalized to housekeeping genes (*HPRT1* and *RPLP0*) in BJ cells transfected with the indicated siRNAs (Control or si*MIR31HG1-2*), treated with DMSO (Control) or 500 nM doxorubicin for 48 h. The graphs show results compared to control DMSO-treated set to 1 (n=3). All statistical significances were calculated using two-tailed Student t-tests. * $p < 0.05$; ** $p < 0.01$; error bars represent mean \pm s.d.

Minor points:

- Formaldehyde-induced crosslinking followed by ribonucleoprotein immunoprecipitation is best abbreviated CLIP (not RIP)

This has been changed.

- Further discussion on how *MIR31HG* might promote the interaction between RSK and RBX1 would be helpful.

Although in vitro kinase assays do not provide physiological conditions, the results suggest that *MIR31HG* interaction with YBX1 might infer an optimal conformation to facilitate its phosphorylation. We have included these new conclusions in the discussion in page 20.

Reviewer #3 (Remarks to the Author):

The manuscript by Marta Montes and Professor Anders H Lund's team focused on the study of lncRNA *MIR31HG* to understand how it regulates the senescence associated secretory phenotype. This work is a continue study after they published their work about lncRNA *MIR31HG* in 2015. Here they included secretome and RNA-binding proteomics to identify the key players with lncRNA *MIR31HG*. This work is interesting and provides the useful information, however, some issues need to be confirmed and clarified as suggested below before publication.

1. Page 5, please change "TGCA data" to "TCGA data".

This has been changed.

2. The top 1 and 2 of GO categories in Supplementary Fig. 1A are Receptor activity and microtubule binding, did the authors further validate the functions? Why did the authors choose not the top 5 functions as the further study?

We have not further validated any of the other functions identified by GO analysis. We agree that there are several additional avenues to pursue based on this data. We here have focused on the SASP components due to its implication in senescence and as a follow up study from our previous *MIR31HG* publication.

3. What is the expression level of ICAM1 after *MIR31HG* knock-down in TIG3 ER:BRAF cell line?

In Figure 1e we showed that ICAM1 and IL1A mRNA levels do not change upon *MIR31HG* depletion in senescent cells. We have now measured the protein levels by western blot and we showed no changes in protein expression. These results are important to show that translation regulation seems to be specific for IL1A. This data is shown in new Supplementary Fig. 3h,i.

Left, Western blot analysis of ICAM1 protein in control or *MIR31HG*-depleted cells. Right, quantification of the density of the bands related to GAPDH from 3 independent experiments.

4. I suggest the authors deposit the secretome and RNA-binding proteome data as well as transcriptome into public proteome and transcriptome databases such as ProteomeXchange and Gene Expression Omnibus (GEO).

All the data generated for this manuscript has been already uploaded to public repositories and is stated in the original version of the manuscript.

“Data availability

Raw data for the RNA-seq are deposited to GEO (GSE144752). Raw data for proteomics experiments are deposited to PRIDE ProteomeXchange (PXD017475).

5. In page 6, The sentence described the results “Among the less secreted proteins upon *MIR31HG* depletion were inflammatory SASP factors IL6 and CXCL1 (Figure 1F, and Supplementary Fig.1D and 1E).”, however, no CXCL1 data was shown in Figure 1H, Supplementary Fig.1D and 1E. How were the validation results using qRT-PCR and western blot analyses? I think the conclusion “*MIR31HG* knock-down reduces the production of several SASP components during OIS already at the transcriptional level and these changes are reflected in the secretome.” is too strong since only

IL6 has the consistent results in different assays. I suggest the author need to provide some evidence to strengthen the conclusions.

To provide further evidence of a decreased SASP secretion we have analysed by ELISA the amount of IL8 and MMP3 in BJ ER:BRAF cells transfected with the indicated siRNAs (Control or siMIR31HG, 1-2), treated with ethanol (Control) or 1 μ M 4-OHT for 72 h. We have observed that the secretion of both components of the SASP was decreased upon MIR31HG knock-down in senescent cells (Fig. A and B). CXCL1 secretion was already validated by ELISA in the original version of the manuscript (see manuscript Figure 1g and rebuttal Fig. C). We tried to detect CXCL1 by western blot of precipitated proteins from the CM, as we did for IL6, without success. However, due to the high sensitivity of the ELISA assay we were able to detect CXCL1 and other SASP components that were undetected by western blot. These new results have been incorporated to the main text in page 6 and are presented in the main Figure 1g. The validation of IL6 by western blot has been moved from Figure 1h to Supplementary Fig. 1I to avoid confusion.

BJ ER:BRAF (Control or siMIR31HG, 1-2) were treated for 72 h with ethanol (Control) or 1 μ M 4-OHT. The CM was then harvested and secreted IL8 (A), MMP3 (B) and CXCL1(C) were measured by ELISA. The graphs show the pg/ml of secreted protein (n>3). All statistical significances were calculated using two-tailed Student t-tests, *p < 0.05; **p < 0.01; error bars represent mean \pm s.d.

6. I suggest the authors showed the bar chart for Figure 2A and 2B.

Quantifications have been made and included in Figure 2.

7. The authors described “NF-kB translocation to the nucleus was inhibited upon MIR31HG depletion (Figure 3C).”, however, I could not see the conclusion in the figure. In Figure 3C, I only see the NF-kB subunit RELA in MIR31HG depletion cells was distributed strongly in the cytosol compared to Control. I suggest the authors use cellular fractionation method to isolate the nuclear and cytosol proteins and further performed Western blot to see whether NF-kB level in nucleus is higher than in cytosol.

In order to verify that NF- κ B is translocated to the nucleus and therefore transcriptionally active in senescence, we used a luciferase reporter plasmid containing 4 NF- κ B binding sites (Addgene Plasmid #111216). As expected, luciferase expression was increased in senescence, but no longer activated upon depletion of *MIR31HG* with 2 different siRNAs (Figure below). As a negative control, a basic promoter plasmid (Promega) was used. These results have been included in Supplementary Fig. 3b.

Luciferase expression in BJ-BRAF cells transfected with luciferase reporter constructs pGL3-promoter (Control) or 4 NF- κ B-luc and with the indicated siRNAs in Control or 4-OHT-treated cells. The graph shows the firefly luciferase relative to renilla units from 6 experiments. All statistical significances were calculated using two-tailed Student t-tests, * $p < 0.05$; ** $p < 0.01$. All error bars represent means \pm s.d.

8. In page 9, the authors described “*MIR31HG* knock-down does not affect mTOR signalling activation (Supplementary Fig. 3H)”, in the data, they didn’t detect the protein level of mTOR, why did they conclude that?

We have analysed phosphorylated p70S6K and phosphorylated S6K, downstream targets of mTOR. We found no changes upon depletion of *MIR31HG* and therefore we concluded that no differences were detected in signalling pathways downstream mTORC.

9. In page 10, the authors described “*IL1A* protein levels were strongly reduced (Figure 4F) whereas the mRNA level remained unaltered (Figure 4D).”, I am curious how to explain the inconsistent results?

The results are not inconsistent. We explained this by showing that *MIR31HG* knock-down reduces *IL1A* translation. We demonstrated this by the reduced levels of *IL1A* mRNA associated to the heavy fractions of the polysome profile (Figure 3I). We have now included a western blot showing reduced nascent *IL1A* synthesis in *MIR31HG* depleted senescent cells. We show this new data in Figure 3m and in point number 7 of this rebuttal.

10. I suggest the authors use some prediction tools to predict the predicted binding motif of interacting proteins on MIR31HG sequence to see whether the proteins identified by mass spectrometry have direct binding sites with MIR31HG. Additionally, I am curious where the binding sites are located in the MIR31HG-YBX1 interaction.

Based on iCLIP studies (Goodarzi, Liu et al. 2015, Wu, Fu et al. 2015), we performed an *in silico* prediction of YBX1 binding sites on *MIR31HG* sequence and identified 5 putative binding sites (BS) (Fig. A and B). We generated *MIR31HG* mutants harbouring deletions including the predicted binding sites (Fig. C). To identify functional YBX1 binding sites on *MIR31HG*, we performed electrophoretic mobility-shift assay (EMSA) experiments incubating the corresponding *in vitro* transcribed RNA with recombinant YBX1. The full length *MIR31HG* (WT) resulted in a band shift when incubated with increasing amounts of YBX1 indicating the binding, whereas *ND4*, a mitochondrial transcript previously used as a negative control (manuscript, Figure 4h) did not bind (Fig. D). Mutant 1 and 5 (Mut1 and Mut5) bind YBX1 to the same extent as the WT whereas mutants 2, 3 and 4 (Mut2, Mut3 and Mut4) did not bind (Fig. D). These results indicate that the predicted BS3 is the responsible for the binding of YBX1 to *MIR31HG*. To validate these results, we performed competition assays incubating ³²P-labelled *MIR31HG* WT transcript with increasing amounts of unlabelled Mut5 or Mut3. We observed that only Mut5 competed for the binding of YBX1 (Fig. E). These results have been incorporated to the manuscript in page 12 and to main Figure 4 (i-k).

Fig. 4. (A) Binding motifs found in published iCLIP data. (B) Representation of the *MIR31HG* sequence with putative YBX1 binding sites (BS). Putative BS of YBX1 from iCLIP data were mapped on *MIR31HG* RNA sequence. BS1-3 were found in Goodarzi *et al.* iCLIP data and BS5-4 were retrieved from Wu *et al.* dataset. (C) Representation of *MIR31HG* truncations (mut1 to mut5) where different BS were removed from the RNA sequence. (D) Electrophoretic mobility assay showing the *in vitro* binding of 2 nM of the corresponding radiolabelled transcript and the indicated amounts of recombinant YBX1. BSA at the highest concentration (400ng) was used as a negative control. (E) Electrophoretic mobility assay showing the *in vitro* binding of 400 ng of YBX1 to 2 nM

of the radiolabelled *MIR31HG* WT in competition with the indicating amounts of unlabelled Mut1 or Mut3.

11. When I compared the data in the manuscript with their previous published paper “Montes, M. et al. The lncRNA *MIR31HG* regulates p16(INK4A) expression to modulate senescence. *Nature communications* 6, 6967 (2015).”, the *IL1A* expression level has no significant change in the manuscript (Supp. Figure 1B), however, has significant change in the previous paper (Figure 3C). How to explain the inconsistent results?

The results are not inconsistent and the observed differences arise as one measurement is from proliferating cells and the other from cells undergoing BRAF-induced senescence. In Figure 3c of our previous manuscript (Montes, M. et al. 2015) we show the levels of *IL1A* after depletion of *MIR31HG* for 72 h in proliferating cells. The graph also shows the levels of *IL1A* mRNA after 4-OHT treatment. What we wanted to claim in this graph was that *MIR31HG* knock-down induced senescence in proliferating cells without the expression of a SASP, which has been described for p16-dependent senescence phenotype (Coppe, Rodier et al. 2011). What we show in the current manuscript in Supplementary Fig. 1b is the effect on *IL1A* mRNA after depletion of *MIR31HG* in senescence conditions.

12. In the MATERIALS AND METHODS section “*MIR31HG* expression analysis in tumor samples”, I suggest the authors provide the data numbers they used.

These numbers have been included.

13. The authors must correct the words and insert space between numbers and units such as “200ug/mL”, “1mg/mL”, “15mL” into “200 ug/mL”, “1 mg/mL”, 15 mL”...

These have been corrected.

Reviewer #4 (Remarks to the Author):

Montes and colleagues show that *MIR31HG* regulates the SASP during BRAF-induced senescence by facilitating YBX1 phosphorylation at serine 102 by the kinase RSK. They connect this event to *IL1A* translation which is a well-known regulator of the SASP. This is an important and novel study that uncovers a new regulatory axis for the SASP. However, the results supporting a translational control are not strong at this point. The following points require authors clarification.

1- In Figure 2a, the induction of p53 is not well characterized. P53 target genes should be part of this analysis.

To further assess the paracrine effect of *MIR31HG*-depleted cells on cell growth we measured the proliferation of BJ and IMR90 fibroblasts that had been in contact with the conditioned media (CM) from BJ ER:BRAF cells (Control or si*MIR31HG*1-2) treated with ethanol (Control) or 1 μ M 4-OHT for 72 h. BJ or IMR90 cells growing in the CM from senescence-treated cells proliferated less than cells in contact with CM from control cells, independently of the presence or absence of *MIR31HG* (Fig. A and B). Furthermore, we observed a lower gene expression of a subset of proliferation-related

genes in BJ and IMR90 cultured in CM from senescent cells (Fig. C and D). BJ cells in contact with CM from 4-OHT treated cells showed an increase p53 protein levels. This increase was observed in *MIR31HG*-depleted cell as well (manuscript, Figure 2a). To validate this result in another cell line, we analysed p53 protein expression in IMR90 cells in contact with the CM described above. As we have previously seen for BJ cells, p53 was increased in all the senescence conditions but not in control-treated cells, independently of *MIR31HG* expression (Fig. E). The expression level of a subset of p53- target genes (Fischer 2017) was also upregulated in BJ and IMR90 cultured in CM from senescent cells in the presence and absence of *MIR31HG* (Fig. F and Fig. G), corroborating the upregulation of the p53 response. These results have been incorporated to the manuscript in page 7. The relative growth in BJ cells is now shown in main Figure 2d. The rest of the data is presented in Supplementary Fig. 2 (a-f).

(A-B) Wild type BJ (A) and IMR90 (B) cells were incubated for 72 h with the CM collected from BJ ER:BRAF cells (Control or si*MIR31HG*1-2) treated with ethanol (Control) or 1 μ M 4-OHT for 72 h. The number of cells was measured by crystal violet staining dissolved in acetic acid and measure at 590 nm. The graph shows the absorbance relative to the control cells set as 1 (n=3). **(C-D)** qRT-PCR analysis of a subset of cell-cycle related genes normalized to housekeeping genes (*HPRT1* and *RPLP0*) in total RNA extracted from cells described in (A-B). The graph shows the RNA expression relative to control cells set to 1 (n=3). **(E)** IMR90 cells were incubated for 72 h with the CM collected from BJ ER:BRAF cells (Control or si*MIR31HG*1-2) treated with ethanol (Control) or 1 μ M 4-OHT for 72 h. Left, whole cell extracts were analysed by western blot for p53 and Vinculin. Molecular weight marker is shown in kDa. Right, quantification of the intensity of the bands using Fiji software

(n=3). (F-G) qRT-PCR analysis of a subset of p53-target genes normalized to housekeeping genes (*HPRT1* and *RPLP0*) in total RNA extracted from cells described in (A-B). All statistical significances were calculated using two-tailed Student t-tests, *p<0.05; **p<0.01; ***p<0.001. All error bars represent means ± s.d. n=3.

2- I do not think the results presented in Fig 3H and quantified in I-J support the authors claim that IL1A is regulated at the translational level. A 25% reduction at the most cannot explain the dramatic results reported in figure 3E and F. Since mRNA levels do not change other mechanisms should be considered including, protein stability, mRNA localisation and miRNAs.

To validate our polysome profile results (main Figure 3j-l) and confirm that the decrease in IL1A protein is due to a defect in protein synthesis rather than protein stability or miRNA regulation, we performed AHA pulse labelling and coupling of newly synthesized proteins to biotin followed by streptavidin pull-down and immunoblotting. We confirmed that *MIR31HG*-depleted senescent cells contained less newly synthesized IL1A compared to control senescent cells, whereas the amount of GAPDH or Vinculin remained unaltered. We were not able to detect the presence of newly synthesized β -Actin, IL6 or ICAM1 in the immunoprecipitated fractions (data not shown) suggesting that their synthesis process might be longer than the 4 h labelling. We believe these results significantly strengthen our original observations and we have included them in Figure 3m and page 10 in the manuscript.

Western blot analysis of newly synthesized IL1A, Vinculin and GAPDH levels upon depletion of *MIR31HG* purified by AHA pulse-labelling and coupling to biotin followed by streptavidin pull-down. As control, proliferating cells without AHA labelling is shown. HRP-streptavidin shows the uniform labelling of newly synthesized proteins coupled to biotin. A representative experiment is shown ($n = 3$).

3- The biological meaning found in this paper is an inhibition of the SASP and its ability to induce invasion when *MIR31HG* is depleted. To validate the reported mechanisms authors should present

similar biological data when analyzing the role of YBX1 phosphorylation and Rsk in BRAF-induced senescence.

In order to probe that reduced YBX1 phosphorylation is responsible for the diminished SASP-induced invasion observed in *MIR31HG*-depleted senescent cells, we performed transwell invasion assays as described for Fig 2 in the manuscript. The CM from ethanol-treated (control) or 1 μ M 4-OHT (senescence) BJ ER:BRAF transfected with the siRNAs control or siRNAs against RSK, YBX1 and *MIR31HG* were used as chemoattractant in the lower chamber of the transwell for MDA-MB-231 breast cancer cells. After 24 h a higher number of MDA-MB-231 cells exposed to CM from senescent cells invaded through the matrigel membrane compared to control CM. In contrast, the CM of RSK and YBX1 knock-down senescent cells inhibited the invasion (Fig. A and B). As previously shown, CM of *MIR31HG*-depleted cells did not promote paracrine invasion. These results demonstrated that RSK and YBX1 are required for invasion. In order to further analyse the requirement of YBX1 phosphorylation, we performed transwell assays using BJ ER:BRAF cells expressing different YBX1 constructs: a WT YBX1, a mutant that is not able to be phosphorylated at S102 (S102A) and a phosphomimic mutant (S102D). Overexpression of YBX1 increased invasion independently of its phosphorylation status (Fig. 8C and 8D), confirming previous findings of YBX1 being involved in metastasis and invasion (Schitteck, Psenner et al. 2007, Castellana, Aasen et al. 2015, El-Naggar, Veinotte et al. 2015). However, under senescence conditions, overexpression of YBX1-WT promoted invasion in a similar extent to S102D, whereas the invasion was reduced when overexpressing S102A (Fig. C and D). Although endogenous YBX1 has been depleted using siRNAs targeting the 3'UTR, remaining YBX1 might explain the fact that all the constructs showed increased invasion in 4-OHT treated conditions compared to control. However, altogether these results suggest a role of YBX1 phosphorylation in promoting invasion. These results have been included in the manuscript (page 14-15) and are shown in Figure 5f,g and Supplementary Fig. 5h,i.

(A) MDA-MB-231 cells were placed in the upper part of a transwell assay in contact with the CM collected from BJ ER:BRAF cells (Control or with the indicated siRNA) treated for 72 h with ethanol (Control) or 1 μ M 4-OHT. After 24 h cells invading the matrigel membrane were stained with crystal violet. Representative images are shown in the figure. Scale bar: 50 μ m. (B) Quantification of the invading cells relative to control ethanol-treated cells (n=3). (C) The CM generated from the different YBX1 phosphomutants was used for transwell assays. Representative images are shown in the figure. Scale bar: 50 μ m. (D) Quantification of the invading cells relative to control ethanol-treated cells (n=2). All statistical significances were calculated using two-tailed Student t-tests, * $p < 0.05$; ** $p < 0.01$. All error bars represent means \pm s.d.

4- Mechanistically figure 5E supports the translational control model. Can the authors repeat their polysome profile analysis in this setting? Also the identification of RNA motifs mediating the regulation will further clarify the translational control model.

The polysome profile analysis in this setting was not possible. The high endogenous levels of YBX1 makes it difficult to be able to distinguish the effect of the YBX1 mutants.

We have been able to identify the RNA motifs in YBX1 responsible for the binding. Based on iCLIP studies (Goodarzi, Liu et al. 2015, Wu, Fu et al. 2015), we performed an *in silico* prediction of YBX1 binding sites on *MIR31HG* sequence and identified 5 putative binding sites (BS) (Fig. A and B). We generated *MIR31HG* mutants harbouring deletions including the predicted binding sites (Fig. C). To identify functional YBX1 binding sites on *MIR31HG*, we performed electrophoretic mobility-shift assay (EMSA) experiments incubating the corresponding *in vitro* transcribed RNA with recombinant YBX1. The full length *MIR31HG* (WT) resulted in a band shift when incubated with increasing amounts of YBX1 indicating the binding, whereas *ND4*, a mitochondrial transcript previously used as a negative control (manuscript, Figure 4h) did not bind (Fig. D). Mutant 1 and 5 (Mut1 and Mut5) bind YBX1 to the same extent as the WT whereas mutants 2, 3 and 4 (Mut2, Mut3 and Mut4) did not bind (Fig. D). These results indicate that the predicted BS3 is the responsible for the binding of YBX1 to *MIR31HG*. To validate these results, we performed competition assays incubating ³²P-labelled *MIR31HG* WT transcript with increasing amounts of unlabelled Mut5 or Mut3. We observed that only Mut5 competed for the binding of YBX1 (Fig. E). These results have been incorporated to the manuscript in page 12 and to main Figure 4 (i-k).

(A) Binding motifs found in published iCLIP data. **(B)** Representation of the *MIR31HG* sequence with putative YBX1 binding sites (BS). Putative BS of YBX1 from iCLIP data were mapped on *MIR31HG* RNA sequence. BS1-3 were found in Goodarzi *et al.* iCLIP data and BS5-4 were retrieved from Wu *et al.* dataset. **(C)** Representation of *MIR31HG* truncations (mut1 to mut5) where different BS were removed from the RNA sequence. **(D)** Electrophoretic mobility assay showing the in vitro binding of 2 nM of the corresponding radiolabelled transcript and the indicated amounts of recombinant YBX1. BSA at the highest concentration (400ng) was used as a negative control. **(E)** Electrophoretic mobility assay showing the in vitro binding of 400 ng of YBX1 to 2 nM of the radiolabelled *MIR31HG* WT in competition with the indicating amounts of unlabelled Mut1 or Mut3.

Minor points:

1- Line 50 : ... and the presence OF senescence-associated heterochromatin...

This has been changed.

2- Figure 2B needs quantification and a magnification bar.

Quantifications and scale bars have been included in the figure.

3- Wajapeyee and al. (Cell, 2008) have shown that IGFBP7 was induced and secreted following BRAFV600E expression and that it could induce senescence and apoptosis in an autocrine/paracrine fashion. Their results show that IGFBP7 is not induced upon BRAFV600E induction (Fig. 1D and fig. 4C), can you comment on that? Also, since IGFBP7 is known to be involved in paracrine senescence, and that when you knock down *MIR31HG* you induce IGFBP7 at the mRNA level (Fig. 1D and fig. 4C) is it more secreted? If so, could it have an impact on the invasion phenotype in figure 2.

The secretome data (Supplementary Table S2) shows a significantly decreased secretion of IGFBP7 in senescence compared to control cells. However, there is no difference between senescence and

senescence *MIR31HG*-KD cells suggesting that IGFBP7 is not likely to be the responsible for the invasion phenotype.

Gene	FC Control vs Sen	p val Control vs Sen	FC siMIR31HG_vs_Sen	p val Sen siMIR31HG_vs_Sen
IGFBP7	1,62	0,0000509396830763681	0,0317	0,894333519227196

4- No scale bars: Fig. 3C, 3D and Fig. 6B

Scale bars have been included.

5- Fig. 6B: can you provide a control for the PLA using only one of the primary antibodies and both probe-conjugated secondary antibodies. These controls ensure that the signal is really due to the binding of the two antibodies in proximity.

We have already performed these controls (No antibody, only RSK and only YBX1 primary antibodies). We did not show the images in the figure but the quantification is shown on the graph on the right.

REFERENCES

- Carlevaro-Fita, J., A. Rahim, R. Guigo, L. A. Vardy and R. Johnson (2016). "Cytoplasmic long noncoding RNAs are frequently bound to and degraded at ribosomes in human cells." *RNA* **22**(6): 867-882.
- Castellana, B., T. Aasen, G. Moreno-Bueno, S. E. Dunn and S. Ramon y Cajal (2015). "Interplay between YB-1 and IL-6 promotes the metastatic phenotype in breast cancer cells." *Oncotarget* **6**(35): 38239-38256.
- Coppe, J. P., F. Rodier, C. K. Patil, A. Freund, P. Y. Desprez and J. Campisi (2011). "Tumor suppressor and aging biomarker p16(INK4a) induces cellular senescence without the associated inflammatory secretory phenotype." *J Biol Chem* **286**(42): 36396-36403.
- El-Naggar, A. M., C. J. Veinotte, H. Cheng, T. G. Grunewald, G. L. Negri, S. P. Somasekharan, D. P. Corkery, F. Tirode, J. Mathers, D. Khan, A. H. Kyle, J. H. Baker, N. E. LePard, S. McKinney, S. Hajee, M. Bosiljic, G. Leprivier, C. E. Tognon, A. I. Minchinton, K. L. Bennewith, O. Delattre, Y. Wang, G. Dellaire, J. N. Berman and P. H. Sorensen (2015). "Translational Activation of HIF1alpha by YB-1 Promotes Sarcoma Metastasis." *Cancer Cell* **27**(5): 682-697.
- Fischer, M. (2017). "Census and evaluation of p53 target genes." *Oncogene* **36**(28): 3943-3956.
- Goodarzi, H., X. Liu, H. C. Nguyen, S. Zhang, L. Fish and S. F. Tavazoie (2015). "Endogenous tRNA-Derived Fragments Suppress Breast Cancer Progression via YBX1 Displacement." *Cell* **161**(4): 790-802.

Roux, P. P., S. A. Richards and J. Blenis (2003). "Phosphorylation of p90 ribosomal S6 kinase (RSK) regulates extracellular signal-regulated kinase docking and RSK activity." Mol Cell Biol **23**(14): 4796-4804.

Schitteck, B., K. Psenner, B. Sauer, F. Meier, T. Iftner and C. Garbe (2007). "The increased expression of Y box-binding protein 1 in melanoma stimulates proliferation and tumor invasion, antagonizes apoptosis and enhances chemoresistance." Int J Cancer **120**(10): 2110-2118.

Skabkin, M. A., O. I. Kiselyova, K. G. Chernov, A. V. Sorokin, E. V. Dubrovin, I. V. Yaminsky, V. D. Vasiliev and L. P. Ovchinnikov (2004). "Structural organization of mRNA complexes with major core mRNP protein YB-1." Nucleic Acids Res **32**(18): 5621-5635.

Stratford, A. L., C. J. Fry, C. Desilets, A. H. Davies, Y. Y. Cho, Y. Li, Z. Dong, I. M. Berquin, P. P. Roux and S. E. Dunn (2008). "Y-box binding protein-1 serine 102 is a downstream target of p90 ribosomal S6 kinase in basal-like breast cancer cells." Breast Cancer Res **10**(6): R99.

Tanaka, T., S. Ohashi, T. Funakoshi and S. Kobayashi (2010). "YB-1 binds to GluR2 mRNA and CaM1 mRNA in the brain and regulates their translational levels in an activity-dependent manner." Cell Mol Neurobiol **30**(7): 1089-1100.

Wu, S. L., X. Fu, J. Huang, T. T. Jia, F. Y. Zong, S. R. Mu, H. Zhu, Y. Yan, S. Qiu, Q. Wu, W. Yan, Y. Peng, J. Chen and J. Hui (2015). "Genome-wide analysis of YB-1-RNA interactions reveals a novel role of YB-1 in miRNA processing in glioblastoma multiforme." Nucleic Acids Res **43**(17): 8516-8528.

REVIEWER COMMENTS

Reviewer #1 (Remarks to the Author):

The authors have answered some of my comments. I appreciate the extensive efforts that the authors have taken to address the concerns raised. However, several questions remain unanswered.

1. For my point 1, I suggested that the authors should determine the full length of MIR31HG and its possible isoforms, but the authors ignored this point in their revision. More importantly, the authors did not exclude the possibility regarding the coding capability of MIR31HG. Given they found the presence of MIR31HG in ribosomes, the authors should design experiments to exclude the possibility that MIR31HG is able to be translated into micropeptides. Specific methods may be referred to in Huang 's article cited below [1].

2. For point 7, the authors failed to present the localization of MIR31HG using RNA-FISH. It is a powerful but not complicated technique. Here, the authors can refer to Clemson's article[2]. This visual image is very important for verifying the localization of MIR31HG in cells.

3. For point 15, authors should show the expression of YBX and RSK.

4. For point 17, authors claim that using bacterial system does not allow them to modulate the expression of MIR31HG. Alternatively, the authors could use the mammalian Two-Hybrid system (e.g. from Promega). The method used can be found in Sun's article [3].

Finally, given the increasing scrutiny of lncRNA related publications, the MIR31HG copy number data are necessary to ensure that these mechanisms are plausible. No such data are presented for the copy number of MIR31HG in the studied cell lines. However, mechanisms of action of MIR31HG would be dependent on an appropriate stoichiometry between MIR31HG and the interacting molecules.

The authors need to address these issues before their article can be considered for publication.

Reference

[1] Huang J Z, Chen M, Chen D, et al. A peptide encoded by a putative lncRNA HOXB-AS3 suppresses colon cancer growth[J]. *Molecular cell*, 2017, 68(1): 171-184. e6.

[2] Clemson C M, Hutchinson J N, Sara S A, et al. An architectural role for a nuclear noncoding RNA: NEAT1 RNA is essential for the structure of paraspeckles[J]. *Molecular cell*, 2009, 33(6): 717-726.

[3] Sun, Xuedan, et al. lncRNA GUARDIN suppresses cellular senescence through a LRP 130-PGC 1 α -FOXO4-p21-dependent signaling axis. *EMBO reports* 21.4 (2020): e48796.

Reviewer #2 (Remarks to the Author):

The authors have addressed all my concerns in full.

Reviewer #3 (Remarks to the Author):

The authors have not only answered my questions and concerns but also improved the manuscript, so I have no further questions.

Reviewer #4 (Remarks to the Author):

The authors have provided new data that address most of my comments and other reviewers' comments. It is still my opinion that this is an important research and I do not have any further comment or criticism.

Response to Reviewer 1 (reviewer comments in blue):

1. For my point 1, I suggested that the authors should determine the full length of MIR31HG and its possible isoforms, but the authors ignored this point in their revision. More importantly, the authors did not exclude the possibility regarding the coding capability of MIR31HG. Given they found the presence of MIR31HG in ribosomes, the authors should design experiments to exclude the possibility that MIR31HG is able to be translated into micropeptides. Specific methods may be referred to in Huang's article cited below [1].

According to our RNAseq data we have based our analysis in the long isoform *NR_027054.2*. The reads map to a long exon 3 (which is not present in *NR_152878.1* and *NR_152879.1*). Also, exon 1 is markedly longer than in isoform *NR_152878.1* (Fig 1A). The same pattern is found in data from ENCODE for fibroblast cell lines BJ and NHLF (Fig 1B). The primers used for qPCR analysis are specific to *NR_027054.2* (Fig 1C). Moreover, the siRNAs used throughout the study target all the 4 isoforms (Fig 1D). Importantly, the *in vitro* experiments and the overexpressing cell lines are created using this sequence. The Materials and Methods section has been updated to specify the focus on *NR_027054.2*.

A

B

C

D

Figure 1. A) Snapshot of IGV browser showing the normalized read counts in the three senescence replicates. Left, exon 1. Right, exons 2 and 3. **B)** Snapshot from UCSC Genome Browser showing reads from RNAseq data available from ENCODE in BJ and NHLF cell lines. **C)** Position of the qPCR primers used in this study. Upper panel, forward primer (red). Bottom panel, reverse primer (red). **D)** position of the siRNAs used in this study (red). Upper panel, siRNA1. Bottom panel, siRNA2.

We have analyzed the coding potential for *NR 027054.2* isoform using different web tools (Figure 2) and all of them showed very limited coding probability. CPAT, 0.029 (Fig 2A), CPC2 0,057 (Fig 2B), portrait, 7,23% (Fig 2C). Moreover, *MIR31HG* sequence is not conserved, particularly in the exons, supporting the non-coding potential (Fig 2D). Importantly, we did not find *MIR31HG* “in” ribosomes, but co-sedimenting with polysomes in a sucrose gradient. Hence, *MIR31HG* may very well be found in a separate complex with similar density.

A CPAT tool

Result for species name : hg19 with job ID :1612261266							
Data ID	Sequence Name	RNA Size	ORF Size	Fickett Score	Hexamer Score	Coding Probability	Coding Label
0	NR_027054.2	2266	246	0.7115	-0.0257085912238	0.029055839582618	no

B CPC2 tool

Summary

Sequence *NR_027054.2* got Fickett score **0.31588** with a **complete** putative ORF **62 AA**, a **pl 8.21319580078**, which, in total, classify it as a **noncoding** sequence with coding probability **0.0569885**.

Details

C PORTRAIT

Sequence name	Coding Probability	Non-Coding Probability
NR_027054.2	7.23%	92.77%

D PhyloP conservation

Figure 2. A-C) Snapshot of the results obtained from the different web browsers to predict coding probability: CPAT (A), CPC2 (B), PORTRAIT (C). **D)** Snapshot from UCSC Genome Browser showing the conservation track for *MIR31HG*.

It is widely accepted that lncRNAs can produce micropeptides and some of these micropeptides have been shown to be functional. The question of a micropeptide would be highly relevant if the study had employed only knockdown and overexpression of the lncRNA. However, here we propose a mechanism based on an RNA:protein interaction that is backed by several techniques and experiments including genetics, proteomics, biochemistry and visualization. This bulk of data cannot be explained by a putative micropeptide.

2. For point 7, the authors failed to present the localization of *MIR31HG* using RNA-FISH. It is a powerful but not complicated technique. Here, the authors can refer to Clemson's article[2]. This visual image is very important for verifying the localization of *MIR31HG* in cells.

We agree RNA-FISH is a powerful method to visualize the localization of a lncRNA. We have put a lot of effort in trying to implement and optimize this technique to our project, with some promising results. Using ViewRNA technology (ThermoFisher) we were able to detect some signal in senescent cells that seems to be specific since it is absent from *MIR31HG* knocked-down cells (Fig 2A). However, no signal is detected in control proliferating cells. We were not able to obtain a reproducible and strong enough signal to be confident with the data, which may reflect reduced accessibility of complexed *MIR31HG* (Fig 2A). Rather than going through multiple rounds of testing and optimizing new probes, we opted for another solid and well-recognized method to obtain the same biological insight. As we answered to reviewer 2, we have analyzed the number of molecules in each compartment upon senescence treatment by cellular fractionation. The results demonstrate that in senescence, *MIR31HG* is mainly located in the cytoplasm, which supports our model (Fig 2B). The fractionation data has been now included in the manuscript in Supplementary Fig 4a.

A

B

Figure 3. A) Representative RNA-FISH images using *MIR31HG* specific probes (green) in control, senescence and senescence *MIR31HG* KD. A mouse RNA sequence was used as negative control, which was not detected (red). **B)** Number of molecules of *MIR31HG* after cellular fractionation calculated by q-PCR using an *in vitro* transcribed *MIR31HG* as template for a standard curve.

3. For point 15, authors should show the expression of YBX and RSK.

We have re-run the samples providing total YBX1 and total RSK.

The levels of p-YBX1 and p-RSK are very low in control proliferating cells. When we deplete RSK with siRNAs we can observe that also p-YBX1 levels are decreased indicating that some degree of regulation might occur in control cells. However, total YBX1 levels seem to be also reduced. Further experiments would be needed to address the role of RSK in proliferating cells.

4. For point 17, authors claim that using bacterial system does not allow them to modulate the expression of *MIR31HG*. Alternatively, the authors could use the mammalian Two-Hybrid system (e.g. from Promega). The method used can be found in Sun's article [3].

We agree that mammalian Two-Hybrid system would have been an interesting method to address the interaction between YBX1 and RSK *in vitro*. Alternatively, we have decided to perform *in vitro* kinase experiments to demonstrate that YBX1 and RSK interact. Moreover, we have observed that the phosphorylation of YBX1 by RSK is modulated by the addition of full length *MIR31HG*. These results complement the PLA experiment where we demonstrate that *MIR31HG* modulates YBX1-RSK interaction *in vivo*. *MIR31HG* regulation of YBX1 phosphorylation is supported throughout the manuscript and further strengthened in the revised manuscript with the functional experiments using *MIR31HG* deletion mutants.

Finally, given the increasing scrutiny of lncRNA related publications, the *MIR31HG* copy number data are necessary to ensure that these mechanisms are plausible. No such data are presented for the copy number of *MIR31HG* in the studied cell lines. However, mechanisms of action of *MIR31HG* would be dependent on an appropriate stoichiometry between *MIR31HG* and the interacting molecules. The authors need to address these issues before their article can be considered for publication.

We have determined the number of molecules in the nucleus and in the cytoplasm in control and senescence by cellular fractionation (see Fig 3B above). The results demonstrate that in senescence, *MIR31HG* is mainly located in the cytoplasm, which supports our model (Fig 2B). *MIR31HG* is very lowly expressed compared to the high levels of YBX1. In contrast to mechanisms where lncRNAs perform “sponging” functions, for instance for miRNAs, our data suggest that the binding of *MIR31HG* to YBX1 acts like a “platform” that allows RSK to phosphorylate YBX1. We speculate this to occur in a processive manner that would not require equimolar amounts of the interactors. This data has been included in the manuscript in Supplementary Fig 4a.

REVIEWERS' COMMENTS

Reviewer #1 (Remarks to the Author):

The authors have addressed most of my comments. I appreciate the authors' extensive efforts. The new data furthermore strengthen the conclusions and therefore publication is warranted.